# Treating Adversarial and Natural Noise Equally: Building a Robust Network for Unified Resilience

## Abstract

The susceptibility of deep recognition algorithms to image degradation exacerbates the disparity between their performance and the resilience of human perception. Such degradations can either be deliberately crafted or naturally occurring. Although both categories of corruption yield similar adverse effects, existing literature has typically treated them separately. We contend that addressing these degradations separately is not conducive to the development of a universally secure and robust system. In this research, we address both types of image degradation, referred to as common corruptions and adversarial perturbations, within a unified framework termed the URoNet. The proposed framework encompasses the following aspects: (i) detecting degraded samples, (ii) mitigating the impact of these degradations, and (iii) establishing potential connections between different forms of degradation. This research introduces a universal framework that employs the URoNET to protect deep learning algorithms from both common/natural and adversarial corruptions. We emphasize the significance not only of the degradations themselves but also of their severity, as they can widen variations within and between classes. Extensive experiments are conducted on various datasets and degradation scenarios, encompassing both seen and unseen settings, to illustrate the effectiveness of the proposed framework.

## 1 Introduction

It has been extensively documented that machine learning algorithms lack robustness when it comes to handling common image corruptions and adversarial perturbations, in stark contrast to human perception (Azulay & Weiss, 2018; Recht et al., 2018; Yin et al., 2019a). The prime reason for the vulnerability of machine learning especially deep learning models might be their biases towards shapes, textures, and other image cues including color (Gavrikov & Keuper, 2024; Wang et al., 2020; Agarwal et al., 2022b). Common image corruptions, as highlighted by (Hendrycks & Dietterich, 2019), often result from various environmental factors such as imaging sensors, viewpoints, and lighting conditions. On the other hand, adversarial perturbations are intentional alterations made to images to deceive classification systems. However, literature observed that both forms of degradation have a similar effect on the high and low-frequency features of images (Lukasik et al., 2023; Saikia et al., 2021; Yin et al., 2019b). The correlation between natural corruptions and adversarial perturbations has been recently explored for a unified detection system (Agarwal et al., 2022a;c). However, these preliminary studies have used the ImageNet classifiers for detecting the corruptions and adversarial perturbations without using any attack knowledge and hence not found generalized in handling unseen attacks.

While these degradations, referring to samples purposefully perturbed through either common corruptions or adversarial perturbations to deceive deep neural networks, stem from entirely distinct mechanisms, distinguishing the added noise patterns can be a formidable challenge through visual inspection alone. In Figure 1, we make a case for not distinguishing natural corruptions and adversarial perturbations when developing a defense mechanism. The noise patterns displayed in the image are extracted from five images within the CIFAR-10 dataset (Krizhevsky et al., 2009) and correspond to two different types of common corruptions and two distinct algorithms for generating adversarial perturbations. It is arduous to discern which pattern belongs to which category solely by examining the noise patterns. Referring to Figure 1, patterns A and D

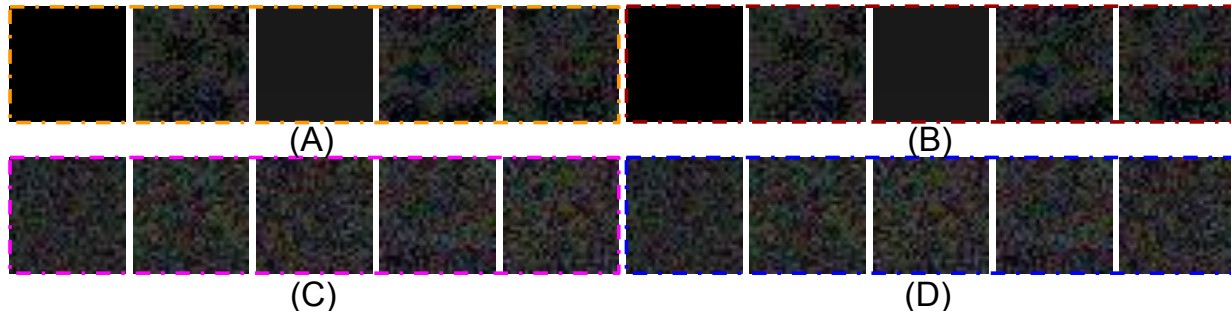

Figure 1: Four different image degradation patterns are obtained using the images of the CIFAR-10 dataset. The degradations are broadly categorized as common corruptions and adversarial perturbations. Can you identify which degradation patterns belong to which category? Each noise pattern is amplified by a factor of 10 for better visibility. Hint: Each row is one degradation type.

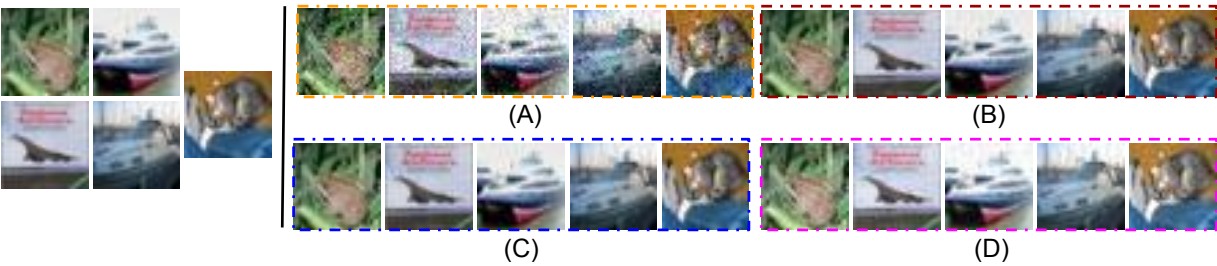

Figure 2: The noisy images generated using the common corruptions and adversarial perturbations. The image at the leftmost side represents the five different clean images. A, B, C, and D are their degraded versions.

correspond to natural corruptions, specifically Gaussian noise, and Uniform noise, respectively. Meanwhile, patterns B and C represent deliberate adversarial perturbations, namely the iterative Fast Gradient Sign Method (IFGSM) (Kurakin et al., 2018) and Projected Gradient Descent (PGD) (Madry et al., 2017). Figure 2 showcases the images that have been altered using the corruptions depicted in Figure 1. While the images in *category A* exhibit visually noticeable alterations, human observers can still classify them correctly with relative ease and confidence. Another common corrupted noisy image displayed in category D features nearly imperceptible modifications akin to the adversarial perturbations shown in categories B and C. These examples further underscore the motivation behind not making a distinction between these degradations, as they can contribute to the development of a unified and robust machine learning framework.

In the literature, several defenses against adversarial perturbations are proposed which vary from adversarial training to the development of perturbation detectors (Abusnaina et al., 2021; Andriushchenko & Flammarion, 2020; Deng et al., 2021; Serban et al., 2020; Shafahi et al., 2020; Agarwal et al., 2023). However, the literature of defenses towards common corruptions is shallow. Popular defenses against common corruptions are based on large-scale data augmentation or utilizing a heavy network backbone (Geirhos et al., 2018; Hendrycks et al., 2019; Mahajan et al., 2018; Rusak et al., 2020; Xie et al., 2020). Existing adversarial training and data augmentation methods are computationally expensive and are not found to be generalizable against unseen corruptions or perturbations with acceptable efficacy (Calian et al., 2021; Mintun et al., 2021). Recently, a few works have started framing defenses from the perspective of denoising (Xie et al., 2019; Salman et al., 2020). The significant drawback of existing defenses is that these degradations are dealt with independently and where common corruptions have received due attention too (Hendrycks et al., 2019; Rusak et al., 2020; Serban et al., 2020; Zhang & Li, 2019).

The primary objective of this study is to bridge the existing gap to establish a reliable machine-learning framework that can effectively withstand both types of corruption. In summary, this research offers the

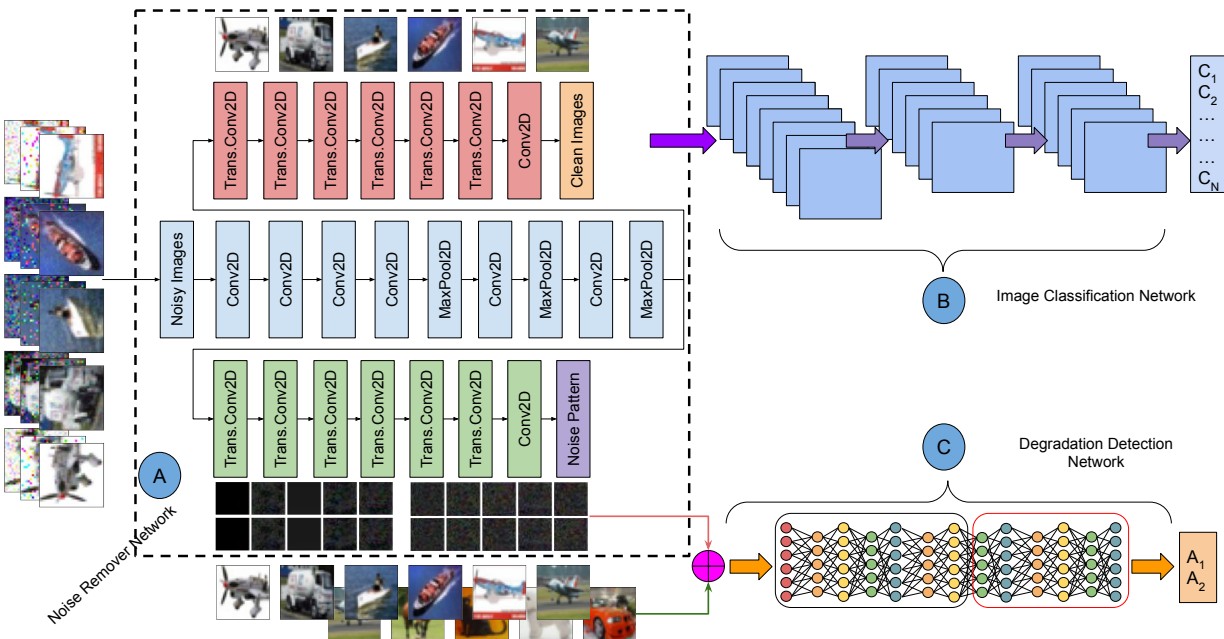

Figure 3: The proposed URoNET framework contains three high-level blocks: (A) aims to remove the noise from the images by breaking the images into two parts: the clean image and noise patterns, and (B) image classification network which utilizes the clean images generated from the noise remover block for enhanced image classification, and (C) degradation detection network aiming to classify the images into binary classes ($A_1$: clean or $A_2$: corrupted). $C_1, C_2, \cdots, C_N$ represent the $N$ classes of the problem. $\bigoplus$ represents data augmentation along the channel dimension where the clean image and noise pattern are concatenated.

following contributions: ❶ proposal of a unified corruption detection and mitigation framework, termed URoNet, capable of effectively addressing both common corruptions and adversarial perturbations, ❷ introduction of a novel image noise removal architecture designed to alleviate the impact of degradations while preserving the performance of Convolutional Neural Networks and ❸ identification of connections between natural corruptions and adversarial perturbations from both detection and mitigation perspectives. These findings have the potential to enhance future research in terms of resilience against both known and unknown perturbations of both types.

## 2 Proposed Unified Security Framework for Degradation Detection and Mitigation

Figure 3 provides an overview of the proposed framework. To achieve our goal of simultaneously handling the detection and mitigation of corruption, we employ a multi-output encoder network in combination with two decoder networks (Mao et al., 2016). One of the decoder networks is dedicated to generating a clean image free from any noise or corruption, while the other aims to isolate and represent the noise pattern itself. The clean image decoded by the first network is subsequently fed into the classifier network to determine the class label for the image. Meanwhile, the noise pattern obtained by the second decoder network serves as auxiliary information for degradation detection. These noise-augmented images, in combination with their clean counterparts, can be used to train a detection network, which constitutes the final component of the framework. This detection network's role is to distinguish between clean and corrupted images. This modular structure allows the framework to adapt to different scenarios by accommodating various combinations of independent networks, ensuring that the system can be tailored to specific requirements. For example, applications working with complex images may opt for a more intricate classification network, while others may opt for a lightweight classifier with fewer parameters to conserve computational resources (Baldock et al., 2021; Jiang et al., 2021).

### 2.1 Training

The comprehensive training structure of the proposed reliable machine learning model, URoNET, can be divided into three main components: (i) The initial component involves training a noise removal network using the degraded images. (ii) An image classification network is trained based on the given number of classes within the dataset. (iii) Finally, an image degradation network is trained to distinguish between corrupted and clean data points. While each of these components can undergo independent training, during the inference stage, they exhibit a degree of interconnection.

To train the noise remover network a deep encoder-decoder is utilized which takes the noisy data as input. To achieve the noisy images, we leverage a set of clean images, say $F$, and obtain a corrupted version of each image in the set as follows:

$$g = f + \eta \tag{1}$$

Here, $\eta$ denotes the noise pattern such as Gaussian or Speckle added to the clean image $f \in F$ to obtain a corrupted image $g$. Then, the autoencoder network (A in Figure 3) is trained using the set of corrupted images $g$ as the input. For each input, the objective of the proposed autoencoder network is to segregate the image into a clean image and noise pattern, such that the loss of the network is defined as:

$$\mu(f', f) + \mu(\eta', \eta) \tag{2}$$

Here, $\mu$ denotes the mean square error function, and $f'$ and $\eta'$ denote the output of the two decoder networks, i.e., the reconstructed clean image and the reconstructed noise pattern, respectively. The benefit of separating clean images and noise patterns arises from the understanding that each noise generation method affects distinct regions of an image owing to the differing intensity of the noise vector applied to the noisy images. Consequently, a straightforward mapping from noisy images to their clean counterparts is insufficient in adequately addressing the diversity in noise characteristics.

Next, we proceed to train the degradation detection network (referred to as C in Figure 3) using the functions $f$ and $g$ for binary classification. However, instead of directly supplying images representing these two classes, we pass the images through the previously trained noise removal network to obtain the potential noise patterns that are added to the images. In the case of images, given that there is natural embedded noise, we can confidently state that the network will not output a blank pattern. Conversely, noisy images will yield a substantial noise vector. Let us denote the obtained noise vectors (images) from the clean and noisy images as $\eta'_f$ and $\eta'_g$ respectively. These vectors are concatenated along the channel dimension to form the corresponding images. If we define the dimensions of the original image as $M * N * c$ where $M$ represents the width, $N$ denotes the height, and $c$ is the number of channels, then the input to the detection network will have dimensions of $M * N * 2c$. In this manner, the detection network can effectively learn to establish a connection between the noise pattern of a corrupted image and the absence of a noise pattern present in a clean image, thereby classifying the input as either clean ($A_1$) or corrupted ($A_2$). The primary objective function of the detection network is to maximize binary classification accuracy.

The final component in the proposed framework is the classifier network (B in Figure 3). This network can undergo separate and independent training, assuming that it will be supplied with clean input data ($f(x, y)$). Its sole focus is on maximizing class-wise accuracy or other relevant metrics typically employed within the specific problem domain, all in the absence of corruption. This distinction sets the proposed framework apart from many other defense mechanisms based on adversarial training. In those approaches, the classifier network must undergo training on both clean and corrupted samples, resulting in a lack of effectiveness when faced with unseen types of corruption and an increase in computational complexity (Geirhos et al., 2018; Kim et al., 2020; Korkmaz, 2021; Mintun et al., 2021; Zhang et al., 2019).

### 2.2 Inference

During the inference phase, an image $h$ is initially fed into the noise removal network, which in turn produces a reconstructed clean image, $h'$, and a reconstructed noise pattern, $\eta'_h$. The clean image $h'$ is subsequently utilized by the classifier network to predict the class label ($C_1, C_2, ..., C_n$). Simultaneously, the degradation detection network receives the channel-wise concatenated image $h' \bigoplus \eta'_h$ as its input. This input is then

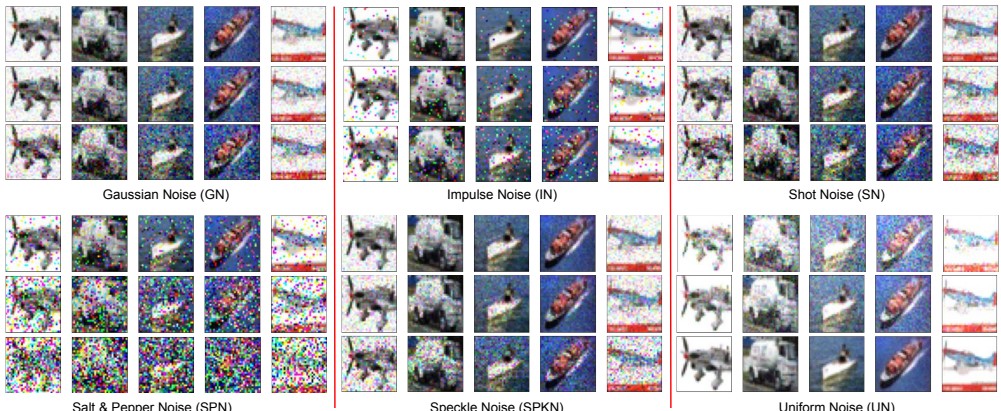

Figure 4: Images are subjected to various common corruptions with differing severity levels, as detailed in Table A1. These variations encompass a range of perceptibility, spanning from readily noticeable to nearly imperceptible.

classified as either clean ($A_1$) or corrupted ($A_2$). The comprehensive end-to-end framework introduced is referred to as "URoNET" designed to prioritize unified robustness in enhancing classification accuracy and enabling effective degradation detection.

### 2.3 Implementation Details

The noise remover encoder-decoder contains six conv layers containing 64, 128, and 256 filters in a pair of layers. ReLU activation is applied at each layer except the last layers utilizing sigmoid activation. The classifier architecture can be any such as on the CIFAR datasets we have used ResNet (He et al., 2016) and Wide-ResNet (Zagoruyko & Komodakis, 2016). For image degradation detection, we have used the Wide-ResNet10-2 architecture. Each component of the framework is trained end-to-end without any pre-training.

**Computational Cost:** The proposed URoNET, designed to offer both detection and mitigation for a range of image degradations, stands out for its computational efficiency. The combined time required for training the proposed noise removal and degradation detection networks is only 25 minutes on a single NVIDIA GeForce RTX 2080 GPU. This level of efficiency positions our algorithm as well-suited for real-world deployment.

## 3 Experimental Results and Analysis

In this research, we have conducted an extensive experimental analysis using multiple popular object recognition datasets, namely CIFAR-10 (Krizhevsky et al., 2009), CIFAR-100 (Krizhevsky et al., 2009), Tiny ImageNet (Le & Yang, 2015), and a subset of ImageNet (Deng et al., 2009). The CIFAR datasets contain pre-defined train and test sets of $50,000$ and $10,000$ images, respectively. We have used six common noise corruptions and generated the noisy images on these datasets with varying levels of severity of the corruptions. The variations in corruption along with their severity help in understanding the impact of the proposed research and establishing its strength in building the universal security framework. The scale parameters of each corruption: Gaussian noise (GN), Uniform noise (UN), Salt and pepper noise (SPN), Shot noise (SN), Impulse noise (IN), and Speckle noise (SPKN) are detailed in Table A1. The corrupted images generated using varying severity levels are also presented in Figure 4. Apart from common noise corruptions, we have used two popular and effective gradient-based adversarial attacks namely projected gradient descent (Madry et al., 2017) and iterative fast gradient sign method (IFGSM) (Kurakin et al., 2018) in zero-shot defense setting.

Table 1: Common corruption detection performance on the CIFAR-10 dataset in terms of average accuracy across severity 1 to 3. The proposed URoNET detection network shows high robustness in handling multiple severities effectively. The lowest severity (i.e., 1) trained detector generalizes well against seen and unseen corruption and severity 2 trained detector yields better average performance the majority of the time. The detailed interpretative caption is given in the text as well. The detailed results are reported in Table B2 of the appendix.

| Train → Test ↓ | GN-1 | GN-2 | UN-1 | UN-2 | SPN-1 | SPN-2 | SN-1 | SN-2 | IN-1 | IN-2 | SPKN-1 | SPKN-2 |
|---|---|---|---|---|---|---|---|---|---|---|---|---|
| GN | 99.97 | **99.98** | 98.89 | **99.98** | **98.77** | 81.71 | **99.99** | 99.86 | 99.99 | **100.00** | 99.96 | **99.98** |
| UN | **96.83** | 89.46 | 83.41 | **96.35** | **80.88** | 64.25 | **87.03** | 81.13 | **94.76** | 84.82 | **95.67** | 84.88 |
| SPN | 82.68 | **99.98** | 99.99 | 99.98 | **100.00** | 100.00 | **99.99** | 99.99 | 99.99 | **100.00** | 99.96 | **99.99** |
| SN | 99.86 | **99.96** | 96.99 | **99.96** | 98.43 | 84.45 | **99.99** | 99.99 | 99.98 | 99.95 | 99.96 | **99.99** |
| IN | 99.87 | **99.98** | 97.87 | **99.98** | 99.58 | 85.90 | **99.68** | 99.50 | 99.99 | **100.00** | 99.96 | 99.86 |
| SPKN | 99.28 | **99.72** | 90.15 | **99.80** | 90.61 | 72.64 | **99.99** | 99.96 | **99.85** | 99.40 | 99.96 | **99.99** |
| Average | 96.42 | 98.18 | 94.55 | **99.34** | 94.71 | 81.49 | 97.78 | 96.74 | 99.09 | 97.36 | 99.25 | 97.45 |

Table 2: Common corruption detection performance on the CIFAR-100 dataset in terms of average accuracy across severity 1 to 3. The proposed URoNET detection network shows high robustness in handling multiple severities effectively. The lowest severity (i.e., 1) trained detector generalizes well against seen and unseen corruption. The detailed interpretative caption is given in the text as well. The detailed results are reported in Table B3 of the appendix.

| Train → Test ↓ | GN-1 | GN-2 | UN-1 | UN-2 | SPN-1 | SPN-2 | SN-1 | SN-2 | IN-1 | IN-2 | SPKN-1 | SPKN-2 |
|---|---|---|---|---|---|---|---|---|---|---|---|---|
| GN | **99.98** | 99.97 | 95.38 | **99.98** | **92.76** | 89.13 | **99.98** | 99.95 | **99.98** | 99.83 | **99.96** | 99.90 |
| UN | **87.54** | 82.16 | 83.43 | **92.22** | 73.36 | **75.58** | 89.38 | **91.84** | **83.19** | 82.01 | 85.47 | **97.33** |
| SPN | 99.98 | **99.99** | 98.65 | **99.11** | **99.99** | 99.92 | 98.89 | 99.95 | **99.99** | 99.98 | 98.76 | **99.90** |
| SN | 99.72 | **99.82** | 93.38 | **99.20** | **93.56** | 91.75 | **99.98** | 99.95 | 99.73 | 99.20 | **99.97** | 99.90 |
| IN | **99.94** | 99.83 | **95.54** | 94.02 | **95.52** | 95.42 | 87.04 | **99.95** | **99.99** | 99.98 | 88.04 | **99.90** |
| SPKN | 97.79 | **98.81** | 86.98 | **96.14** | **83.69** | 83.41 | **99.93** | 99.95 | **98.15** | 95.77 | **99.97** | 99.90 |
| Average | 97.49 | 96.76 | 92.23 | 96.78 | 89.82 | 89.20 | 95.87 | 98.60 | 96.84 | 96.13 | 95.36 | **99.47** |

## 3.1 Detection

First, using CIFAR datasets, we conducted experiments focused on detecting images affected by common corruptions. We introduced corrupted images into both the training and testing sets. While the training set consisted of both corrupted and clean images, which are employed for training the noise removal and detection networks, the testing set is reserved for evaluation. Our results encompass a wide range of scenarios, including those where the network encounters both known (seen) and unknown (unseen) corruption types and varying levels of corruption severity. This extensive testing is crucial for assessing the algorithm's real-world performance in situations where the network may encounter corruption types and severity levels it is not specifically trained.

### 3.1.1 Seen Distribution Detection

The results of common corruption detection on the CIFAR-10 and CIFAR-100 datasets are reported in Table 1 and Table 2, respectively. Here the results are reported regarding average detection accuracy across severties 1 to 3. The analysis can be divided based on corruption type and correction severity. For example, as observed in Table 1, when the Gaussian noise is used in training, it yields at least 99.72% detection accuracy across corruption types except for Uniform noise. However, the Gaussian noise-trained detector can detect 89.46% corrupted images showcasing its generalizability and reflecting that the proposed approach is noise agnostic. A similar observation can be seen from the analysis presented in Table 2, where the Gaussian noise with severity two trained detector can detect unseen noisy images by at least 82.16% average accuracy across three severities where severities one and three are also unseen during training. It shows that the proposed approach can not only handle smaller but also higher severity of noises. In brief, on the CIFAR-10 dataset,

Table 3: Common corruption detection in the out-of-distribution (unseen/cross dataset) scenario. Similar to the seen dataset protocols, the proposed detection network can handle out-of-distribution images whether coming from the same or different corruptions with high efficacy.

| Train ↓ | GN | UN | SPN | SN | IN | SN | GN | UN | SPN | SN | IN | SN |
|---------|------|------|------|------|------|------|------|------|------|------|------|------|
| Test → | Train CIFAR-10 → Test CIFAR-100 | | | | | | Train CIFAR-100 → Test CIFAR-10 | | | | | |
| GN | 99.90 | 90.89 | 99.84 | 99.66 | 99.23 | 98.40 | 99.99 | 67.02 | 99.99 | 99.73 | 99.95 | 96.78 |
| UN | 99.47 | 99.46 | 99.47 | 99.13 | 61.50 | 97.62 | 99.9 | 99.89 | 74.69 | 99.73 | 55.29 | 98.57 |
| SPN | 93.59 | 50.01 | 99.98 | 93.41 | 98.55 | 76.34 | 83.81 | 50.00 | 100 | 86.62 | 87.34 | 68.67 |
| SN | 99.91 | 64.20 | 99.91 | 99.91 | 98.82 | 99.91 | 100 | 77.85 | 99.92 | 100 | 65.10 | 99.99 |
| IN | 99.87 | 82.62 | 99.87 | 99.79 | 99.87 | 99.01 | 99.99 | 52.28 | 100 | 99.74 | 100 | 97.61 |
| SPKN | 99.73 | 87.11 | 99.73 | 99.73 | 99.73 | 99.73 | 99.95 | 68.8 | 99.89 | 99.99 | 68.42 | 99.99 |

Uniform noise and speckle noise with severity two yield almost perfect detection accuracy across corruptions whether they are seen or unseen during training.

The above results reveal that there might be some connection that can be utilized to detect common corruptions in the out-of-distribution images. Out-of-distribution is an interesting scenario (Mandal et al., 2019; Ren et al., 2019; Vyas et al., 2018) where the images of one dataset are used for training and images of other datasets are used for evaluation. However, such a study is not explored for the detection of corruption examples in unseen or out-of-distribution samples. In the case of the proposed research, the importance of such a study can be thought from the following points: the datasets used for corruption belong to object recognition and are of the same resolution ($32 \times 32$), and in the case of zero-shot setting, images of unseen object class might come which might be corrupted. However, the decision of whether they belong to any corrupted category is critical for the right decision.

### 3.1.2 Out-of-Distribution Detection

Towards out-of-distribution corruption examples detection, two-way experiments are performed: (i) training on the CIFAR-10 dataset using each corruption individually and tested on the images of the CIFAR-100 dataset corrupted using seen or unseen corruption of lowest severity used and (ii) the reverse case. When the corruption detection algorithm is trained on the CIFAR-10 dataset and tested on the CIFAR-100 dataset, except in a few cases, the detection algorithms show significantly higher performance (Table 3). The few cases where the performance is lacking are when the detector is trained on the UN corruption and tested on the IN corruption. In another scenario, training on SN and testing on unseen UN attacks shows a low detection performance (64.2%). We want to highlight here that in these cases not only the datasets are different but the corruption type is also different and it reveals the high inter-class variability among the corruptions. Gaussian noise (GN) shows the highest level of generalization both towards out-of-distribution data but also towards unseen corruption and yields at least 90.89% accuracy on the CIFAR-100 images. Speckle noise (SPKN) is found second-best in identifying corrupted images in such wide variations in testing protocol. The effectiveness of GN corruption is also seen under the reverse case, i.e., where the training has been done on CIFAR-100 and testing is performed on the CIFAR-10 dataset.

### 3.1.3 Open-set Adversarial Perturbation Detection

Adversarial perturbations have tremendous success in fooling 'any' deep neural network classifier by being quasi-imperceptible (Agarwal et al., 2020; Hendrycks et al., 2021; Pedraza et al., 2021). Several research works have been proposed to tackle the detection of adversarial examples in generalized settings (Abusnaina et al., 2021; Agarwal et al., 2021a;b; Drenkow et al., 2022; Goswami et al., 2019); however, they are not explored in the context of their connection with common corruptions. Both kinds of degradation have proven effective in fooling CNNs with high confidence. Therefore, to build a universally secure system we tackle both degradations simultaneously. *The first question we ask is whether a detector trained on common corruptions can be effective in detecting adversarial perturbations.* If yes, then is each corruption equally effective for this purpose or does any specific corruption perform better in identifying adversarial

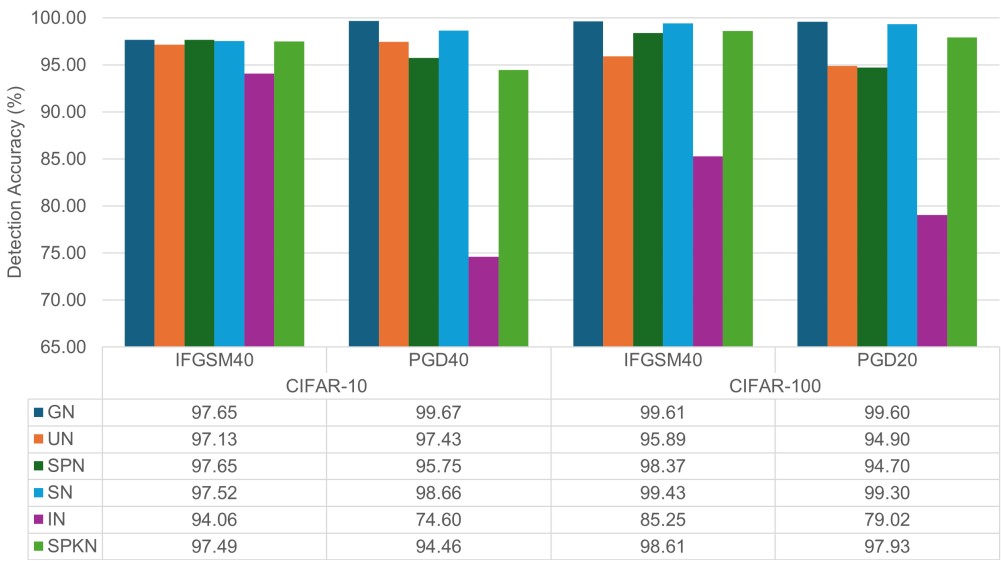

| | IFGSM40 | PGD40 | IFGSM40 | PGD20 |
|---|---|---|---|---|
| | CIFAR-10 | | CIFAR-100 | |
| GN | 97.65 | 99.67 | 99.61 | 99.60 |
| UN | 97.13 | 97.43 | 95.89 | 94.90 |
| SPN | 97.65 | 95.75 | 98.37 | 94.70 |
| SN | 97.52 | 98.66 | 99.43 | 99.30 |
| IN | 94.06 | 74.60 | 85.25 | 79.02 |
| SPKN | 97.49 | 94.46 | 98.61 | 97.93 |

Figure 5: Common corruption detection in terms of average accuracy across severities on the CIFAR-10 and CIFAR-100 datasets where the detectors are trained on FGSM and PGD adversarial perturbations. In contrast to adversarial examples detection trained on common corruption, adversarial detector shows significant success in identifying common corruption. The detailed results are reported in Table B5 of the appendix.

examples? We have generated two highly effective and complex adversarial perturbations namely iterative FGSM (IFGSM) (Kurakin et al., 2018) and PGD (Madry et al., 2017). Interestingly, almost all common corruptions show ineffectiveness in detecting adversarial examples for both datasets. However, surprisingly, uniform noise (UN) with the lowest severity shows a very strong correlation between the image degradation types. When the detector which is trained with the aim of detection of UN examples with severity 0.1 is used to detect the IFGSM perturbations, it yields 82.86% and 87.23% accuracy on the CIFAR-10 and CIFAR-100 datasets, respectively. Moreover, the confidence in adversarial perturbation detection increases significantly when the PGD attack which is considered one of the strongest first-order universal adversaries is brought for detection. On both datasets, the UN-corrupted trained detector yields almost perfect PGD perturbation detection accuracy. The degradation induced by UN corruption with severity 0.1 is found quasi-imperceptible (the third row of UN corrupted examples in Figure 4) similar to the adversarial perturbations. Results of this interesting study are reported in Table B4 in the appendix.

Unlike earlier findings related to cross-degradation detection, it is evident that detectors trained on adversarial perturbations exhibit a significant level of generalization in detecting common corruptions, even when they vary in severity levels. This phenomenon can be attributed to the subtle nature of adversarial perturbations. If a detector can successfully identify such subtle perturbations, it becomes more adept at detecting common corruptions that possess similar or even more substantial perturbations. For instance, when training is conducted using IFGSM degradation on the CIFAR-10 dataset, and testing is performed on common corruptions, the results indicate a detection accuracy of at least 87.02%. However, it is worth noting that on each dataset, the IN corruption with the lowest severity level remains particularly challenging to detect. The high detection robustness of adversarial perturbations against common corruptions is demonstrated in Figure 5.

## 3.2 Mitigation

The second objective of this research is to be able to mitigate the impact of these highly adverse degradations to preserve recognition performance. Figures 6 and 7 show the mitigation results on the CIFAR-10 and CIFAR-100 datasets, respectively.

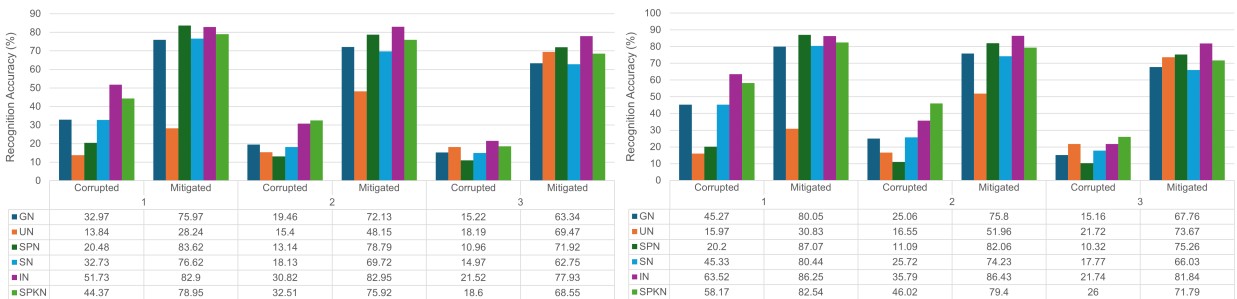

| | Corrupted | Mitigated | Corrupted | Mitigated | Corrupted | Mitigated | | Corrupted | Mitigated | Corrupted | Mitigated | Corrupted | Mitigated |
|---|---|---|---|---|---|---|---|---|---|---|---|---|---|
| | | 1 | | 2 | | 3 | | | 1 | | 2 | | 3 |
| GN | 32.97 | 75.97 | 19.46 | 72.13 | 15.22 | 63.34 | GN | 45.27 | 80.05 | 25.06 | 75.8 | 15.16 | 67.76 |
| UN | 13.84 | 28.24 | 15.4 | 48.15 | 18.19 | 69.47 | UN | 15.97 | 30.83 | 16.55 | 51.96 | 21.72 | 73.67 |
| SPN | 20.48 | 83.62 | 13.14 | 78.79 | 10.96 | 71.92 | SPN | 20.2 | 87.07 | 11.09 | 82.06 | 10.32 | 75.26 |
| SN | 32.73 | 76.62 | 18.13 | 69.72 | 14.97 | 62.75 | SN | 45.33 | 80.44 | 25.72 | 74.23 | 17.77 | 66.03 |
| IN | 51.73 | 82.9 | 30.82 | 82.95 | 21.52 | 77.93 | IN | 63.52 | 86.25 | 35.79 | 86.43 | 21.74 | 81.84 |
| SPKN | 44.37 | 78.95 | 32.51 | 75.92 | 18.6 | 68.55 | SPKN | 58.17 | 82.54 | 46.02 | 79.4 | 26 | 71.79 |

Figure 6: Common corruption mitigation on the CIFAR-10 dataset using ResNet (left) and Wide-ResNet16-8 (right). On clean images, ResNet and WRN yield an accuracy of 91.81% and 93.41%, respectively. In brief, the noise remover trained with severity coming at the test time shows better robustness as compared to training on different severity levels. However, the unseen severity training can boost the performance multiple folds. The detailed interpretative caption is given in the text as well. The detailed results with seen and unseen severity are reported in Table C6 of the appendix.

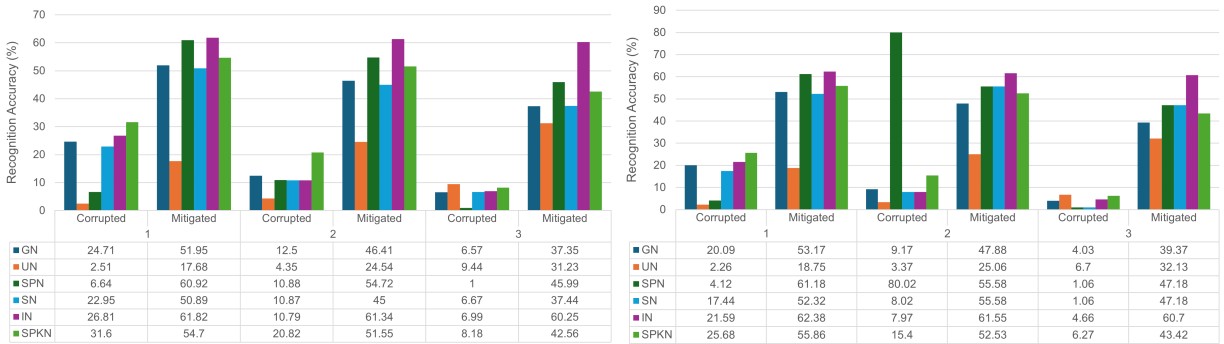

| | Corrupted | Mitigated | Corrupted | Mitigated | Corrupted | Mitigated | | Corrupted | Mitigated | Corrupted | Mitigated | Corrupted | Mitigated |
|---|---|---|---|---|---|---|---|---|---|---|---|---|---|
| | | 1 | | 2 | | 3 | | | 1 | | 2 | | 3 |
| GN | 24.71 | 51.95 | 12.5 | 46.41 | 6.57 | 37.35 | GN | 20.09 | 53.17 | 9.17 | 47.88 | 4.03 | 39.37 |
| UN | 2.51 | 17.68 | 4.35 | 24.54 | 9.44 | 31.23 | UN | 2.26 | 18.75 | 3.37 | 25.06 | 6.7 | 32.13 |
| SPN | 6.64 | 60.92 | 10.88 | 54.72 | 1 | 45.99 | SPN | 4.12 | 61.18 | 80.02 | 55.58 | 1.06 | 47.18 |
| SN | 22.95 | 50.89 | 10.87 | 45 | 6.67 | 37.44 | SN | 17.44 | 52.32 | 8.02 | 55.58 | 1.06 | 47.18 |
| IN | 26.81 | 61.82 | 10.79 | 61.34 | 6.99 | 60.25 | IN | 21.59 | 62.38 | 7.97 | 61.55 | 4.66 | 60.7 |
| SPKN | 31.6 | 54.7 | 20.82 | 51.55 | 8.18 | 42.56 | SPKN | 25.68 | 55.86 | 15.4 | 52.53 | 6.27 | 43.42 |

Figure 7: Common corruption mitigation on the CIFAR-100 dataset using Wide-ResNet28-10 (left) and Wide-ResNet16-8 (right). On clean images, WRN28-10 and WRN16-8 yield an accuracy of 76.21% and 74.57%, respectively. The performance drops under corrupted images and boosts drastically when the proposed noise remover is applied to the images. The detailed results with seen and unseen severity are reported in Table C7 of the appendix.

### 3.2.1 Results on CIFAR Datasets

The ResNet (RN) and Wide-ResNet16-8 (WRN) classifiers yield 91.81% and 93.41% classification accuracy on the clean test images of the CIFAR-10 dataset, respectively. The RN classifier can be seen as a deep classifier whereas, the WRN classifier is both deep and wide. We observe that every corruption even added with the lowest severity can reduce the performance of both classifiers. In terms of corruption, among all the noises, UN corruption is found highly stealthy at the lowest severity level. However, the proposed URoNet drastically boosts the recognition performance across corruption types and their associated intensities (severities). For example, on the UN corrupted images of severities one and three, the performance of the ResNet classifier increases by 14.4% and 51.28%, respectively. A similar observation can be seen across the corruption type, its intensities, and the target classifier. We want to highlight that the proposed URoNet does not utilize any knowledge of the target classifier and hence its effectiveness in defending 'any' target classifier, makes it real-world friendly.

In contrast to CIFAR-10, the impact of noise corruption is significantly higher on the CIFAR-100 dataset, which might be due to a large number of classes leading to higher complexity. SPN corruption is again able to reduce the performance to the lowest accuracy, i.e., 1.0% on WRN16-8 and 1.06% on WRN28-10 which is random chance accuracy for 100 classes. However, again as observed on the CIFAR-10 dataset, the proposed noise remover algorithm can boost the recognition performance substantially. The proposed algorithm can

Table 4: Mitigation results on the CIFAR-10 dataset. The proposed 'URoNet' outperforms traditional denoising (DCAE) and existing algorithms by a significant margin across each corruption. Raw represents the accuracy of the corrupted test set.

| Noise | CNN | Raw | (Salman et al., 2020) | (Xie et al., 2019) | DCAE | URoNet |
|-------|-----|-----|-----------------------|--------------------|------|--------|
| GN | ResNet | 32.97 | 55.65 | 57.21 | 54.21 | **75.97** |
| | WRN | 45.27 | 60.04 | 45.69 | 59.30 | **80.05** |
| IN | ResNet | 51.73 | 58.17 | 65.20 | 59.75 | **82.90** |
| | WRN | 63.52 | 62.34 | 58.36 | 53.87 | **86.25** |
| SN | ResNet | 32.73 | 64.53 | 48.31 | 56.47 | **76.62** |
| | WRN | 45.33 | 58.76 | 46.89 | 60.02 | **80.44** |
| SPKN | ResNet | 44.37 | 55.94 | 71.05 | 57.90 | **78.95** |
| | WRN | 58.17 | 69.93 | 57.01 | 61.82 | **82.54** |
| UN | ResNet | 18.19 | 64.24 | 66.53 | 53.99 | **69.47** |
| | WRN | 21.72 | 67.93 | 61.00 | 58.08 | **73.67** |
| SPN | ResNet | 20.48 | 56.55 | 42.18 | 50.07 | **83.62** |
| | WRN | 20.20 | 60.98 | 35.76 | 54.02 | **87.07** |

Table 5: Comparison results against SOTA works. "Clean" indicates Top-1 clean accuracy (%) (higher is better). "mCE" shows the performance (%) over 15 corruption types (less is better).

| Algorithm | CIFAR10-C | | CIFAR100-C | |
|-----------|-----------|------|------------|------|
| | Clean↑ | mCE↓ | Clean↑ | mCE↓ |
| (Kim et al., 2021) | 75.3 | 45.6 | 57.9 | 48.1 |
| (Yang et al., 2022) | 77.2 | 41.9 | 58.0 | 46.4 |
| (Zou et al., 2020) | 62.3 | 45.1 | 55.2 | 45.8 |
| (Hendrycks et al., 2019) | 79.5 | 43.4 | 60.6 | 44.9 |
| GLOT-DR (Phan et al., 2023) | 83.7 | – | 55.7 | – |
| DuTaNet (Agarwal et al., 2024) | 81.7 | 39.6 | 62.4 | 42.2 |
| **Proposed URoNet** | **83.4** | **39.1** | **63.5** | **41.4** |

increase recognition performance on the CIFAR-100 dataset as well reflecting that it is dataset agnostic. Similar to the improvement on CIFAR-10, the increase in the CIFAR-100 is also manifold. For instance, on the IN corruption with severity three, the proposed algorithm boosts the recognition performance of WRN28-10 from 6.99% to 60.25%. The boost is even higher when the shallow classifier (WRN16-8) is used, i.e., the performance is boosted from 4.66% to 60.7%. We want to mention that the performance improves by more than 39 and 40 times in cases where perturbations can reduce the performance to a random chance level (i.e., 1.0%) demonstrating the strength of the proposed defense.

**Comparison with existing state-of-the-art works:** Extensive experiments performed using multiple datasets and CNN architectures showcase the importance of the proposed noise remover and detection algorithm in building unified security and robustness. To further demonstrate the effectiveness of the proposed work, a comprehensive comparison with existing state-of-the-art (SOTA) algorithms (Salman et al., 2020; Xie et al., 2019) has also been performed. (Salman et al., 2020) utilizes the denoising block which generates multiple copies of noisy images of the input images with multiple loss functions aiming to reduce the noise and improve the classification performance. It is seen in the literature that the feature space of deep networks is perturbed when the noisy images are processed through them and leads to misclassification (Goswami et al., 2019). Hence, in contrast to purifying the input image, (Xie et al., 2019) proposed the purification of features. The comparison has been performed using multiple corruption algorithms that not only reflect the capacity (in increasing the robustness against individual corruptions) but also the generalizability (handling unseen corruptions). As seen each corruption can degrade the recognition performance of the image classification networks. The proposed and existing mitigation algorithms aim to restore recognition accuracy. The results reported in Table 4 show that the existing algorithms and traditional image-denoising architectures miserably failed in doing so. Further, these algorithms are not found generalized in handling corruption and yield poor recognition accuracy. On top of that, the existing algorithms have heavy computational demand

Table 6: PSNR/SSIM (↑) comparison with SOTA.

| Algorithm | CIFAR10-C | | CIFAR100-C | |
|---|---|---|---|---|
| | PSNR | SSIM | PSNR | SSIM |
| (Byun et al., 2021) | 38.7 | 0.91 | 36.1 | 0.84 |
| (Wang et al., 2022) | 35.4 | 0.84 | 32.7 | 0.81 |
| (Zhang et al., 2022a) | 37.5 | 0.86 | 34.3 | 0.82 |
| (Yao et al., 2023) | 39.5 | 0.88 | 33.7 | 0.82 |
| (Zhang et al., 2022b) | 39.2 | 0.90 | 35.1 | 0.84 |
| DuTaNet (Agarwal et al., 2024) | 40.7 | 0.93 | 37.8 | 0.86 |
| **Proposed URoNet** | **42.8** | **0.95** | **39.2** | **0.87** |

to develop an image enhancement algorithm. For instance, (Xie et al., 2019) have utilized the concept of adversarial training and the algorithm took 52 hours on 128 V100 GPUs to defend the PGD attack only.

We have also performed an extensive comparison with recent SOTA image-denoising algorithms and the results are reported in Tables 5 and Table 6. The performance of the proposed algorithm is compared using metrics such as performance on clean images (higher the better), mCE (lower the better), PSNR (higher the better), and SSIM (higher the better), following the protocols of the existing works. (Kim et al., 2021) have used the invertible encoder-decoder architecture, where the first stage learns the object recognition network and the second stage produces low-quality and high-quality (HQ) images. Due to the high dependency on high-quality images, the algorithm is found less resilient across multiple forms of corruption. (Yang et al., 2022) proposed the feature distillation method to produce HQ features. (Phan et al., 2023) have proposed the defense algorithm by simultaneously optimizing the local and global regularization. While GLOT-DR (Phan et al., 2023) yields slightly better performance on the CIFAR10-C dataset, the proposed algorithm outperforms the existing algorithms by a significant margin and is found generalized in handling unseen noise corruptions. Further, apart from these algorithms, we have compared our algorithm (Byun et al., 2021), IDR (Zhang et al., 2022a), (Wang et al., 2022), (Yao et al., 2023), and (Zhang et al., 2022b) in terms of the PSNR metric. (Byun et al., 2021) propose the network to model the Gaussian-Poisson distribution. Along with the requirement of a Gaussian-Poisson pair, the algorithm is computationally heavy having $2,512$ MB as compared to the proposed algorithm. (Yao et al., 2023) have proposed the two-branch self-supervised network, where one branch performs denoising, and another branch modulates denoising. (Zhang et al., 2022b) utilize two different corrupted information such as one short-exposure noisy image and a long-exposure blurry image. The PSNR value of the proposed algorithm is at least 3.3% higher than (Yao et al., 2023) on the CIFAR-10 and CIFAR-100 datasets. A similar improvement can be seen in terms of SSIM values reflecting that the proposed defense can effectively remove the noise from the images.

Apart from the above defense approaches, literature also has algorithms based on the utilization of diffusion models such as DensePure (Xiao et al., 2022) and DiffPure (Nie et al., 2022). The key technical distinction between our proposed approach and DensePure lies in their respective methodologies. DensePure involves multiple denoising runs that use the reverse process of the diffusion model, each with different random seeds, to generate multiple reversed samples. These reversed samples are then passed through the classifier, and the final prediction is made by majority voting of the inferred labels. This approach entails significant computational complexity due to the multiple denoising runs through diffusion and multiple rounds of classification. In contrast, our proposed algorithm achieves adversarial and corruption purification with a single run, making it computationally more efficient. Additionally, our algorithm adopts a multi-task architecture that not only seeks to mitigate the impact of noise, including both corruptions and adversarial perturbations but also classifies the images into clean and noise categories. This classification serves a practical purpose, as it allows for the avoidance of using a purifier when the images are clean, resulting in computational cost savings. This functionality is not available in DensePure. Notably, due to this absence, it is observed that adversarial defense algorithms, mitigate adversarial perturbations to varying degrees but may also reduce the performance of clean images. Moreover, it is important to highlight that DensePure is specifically evaluated for adversarial perturbations and is assessed using Vision Transformers (ViTs). Similarly, DiffPure also employs a computationally intensive process to iteratively purify adversarial examples. It begins with a forward diffusion process using a small amount of noise and subsequently recovers the clean image through a reverse generative process.

Table 7: Common corruption mitigation results on the Tiny ImageNet dataset. N (↓) and P (↑) represent the performance on noisy and mitigated images, respectively. The proposed noise remover is trained on an unseen severity level (i.e., severity 1 which is denoted as -1). Similar to the other datasets, the proposed noise remover can restore the recognition performance of any classifier operating on any large-scale dataset. A few values are highlighted to quickly highlight the strength of the proposed algorithm.

| Train ↓ | DenseNet | | | | MobileNet | | | | VGG16 | | | |
|---|---|---|---|---|---|---|---|---|---|---|---|---|
| Test → | severity 2 | | severity 3 | | severity 2 | | severity 3 | | severity 2 | | severity 3 | |
| | N (↓) | P (↑) | N (↓) | P (↑) | N (↓) | P (↑) | N (↓) | P (↑) | N (↓) | P (↑) | N (↓) | P (↑) |
| GN-1 | 22.43 | 39.16 | 9.33 | 33.89 | 20.60 | 33.37 | 8.18 | 27.06 | 29.23 | 37.43 | 16.19 | 31.95 |
| UN-1 | 5.39 | 23.11 | 13.93 | 33.61 | 4.45 | 17.68 | 11.95 | 27.05 | 9.27 | 21.87 | 20.99 | 30.96 |
| SPN-1 | **0.81** | **39.50** | 0.52 | 16.60 | **0.85** | **33.73** | 0.57 | 13.45 | **1.46** | **36.97** | 0.68 | 17.45 |
| SN-1 | 20.04 | 38.61 | 9.46 | 33.78 | 19.35 | 32.45 | 8.79 | 27.96 | 27.79 | 35.97 | 16.61 | 32.84 |
| IN-1 | 14.67 | 43.99 | **8.20** | **43.90** | 16.69 | 38.26 | **8.87** | **38.00** | 22.80 | 39.47 | **15.06** | **39.12** |
| SPKN-1 | 34.14 | 40.55 | 13.55 | 34.97 | 31.38 | 35.06 | 13.40 | 30.15 | 36.39 | 36.59 | 21.13 | 33.77 |

### 3.2.2 Results on Tiny-ImageNet

We have also demonstrated the strength of the proposed noise remover network by showing that it can scale to a large-scale dataset both in terms of classes and resolution of the images. For this purpose, we have used the **Tiny Imagenet** dataset (Le & Yang, 2015) which contains high-resolution images of size $64 \times 64 \times 3$ of 200 object classes. Not only the dataset, but we have extended the CNN architectures as well to establish the classifier-agnostic nature of the proposed URoNet architecture. We have used *DenseNet-121* (Huang et al., 2017), *MobileNet* (Howard et al., 2017), and *VGG-16* (Simonyan & Zisserman, 2014) pre-trained on the ImageNet and fine-tuned for object classification on Tiny ImageNet training images. DenseNet, MobileNet, and VGG architectures yield 62.08%, 57.53%, and 50.41% classification accuracy on the clean test images of the dataset, respectively. We have evaluated the performance of the proposed noise remover against the unseen severity level of the corruption to check the generalizability. In other words, severity level one is used to train the noise remover network and the testing has been performed on the corrupted images using severity levels two and three.

Among the common corruptions, Salt & Pepper noise (SPN) is identified as the most severe, causing near-complete performance degradation in each network, with an impact close to 0% accuracy. However, the proposed noise removal network plays a crucial role in substantially restoring the classification performance of these networks. For instance, when two images corrupted with SPN at severity level two are used, the performance of DenseNet drops from 62.08% to a mere **0.81**%. However, after applying the mitigation provided by our framework, the performance is restored to **39.50**%. Similarly, the vulnerable performance of the VGG and MobileNet architecture impacted by SPN-2, increases by **35.51**% and **32.88**%, respectively. The demonstrated robustness extends beyond Salt & Pepper noise (SPN) to encompass various other corruptions and their respective severities. This robustness is consistently evident in multiple experiments, reinforcing our claim that the proposed network can be effectively employed with any dataset, type of corruption, or classifier, regardless of whether they are encountered during the training process. The outcomes of this robustness on the Tiny ImageNet dataset are reported in Table 7.

### 3.2.3 Results on ImageNet

To further expand the impact of the proposed URoNet, we have now utilized the high-resolution subset[1] of the *'ImageNet'* dataset (Deng et al., 2009). Similar to the experiments on Tiny ImageNet, we have used three classical CNN architectures. DenseNet, MobileNet, and VGG architecture yield 96.10%, 95.20%, and 92.50% classification accuracy on the clean test images of the subset, respectively. The generalization of the proposed framework is evaluated by training on severity level one corruption and tested on levels two and three. The results of the proposed algorithm on the ImageNet subset are reported in Table 8. Several interesting results are observed in the high-resolution dataset: (i) the proposed noise remover exhibits resolution-agnostic behavior, effectively mitigating the impact of corruption by a substantial margin, and

---

[1]https://github.com/fastai/imagenette

Table 8: Common corruption mitigation results on the subset of **'ImageNet'** dataset (Deng et al., 2009). N (↓) and P (↑) represent the performance on noisy and mitigated images, respectively. The proposed noise remover is trained on an unseen severity level (i.e., severity 1 which is denoted as -1). Similar to multiple datasets, the proposed noise remover can restore the recognition performance of any classification operating on any large-scale dataset. A few values are highlighted to quickly highlight the strength of the proposed algorithm.

| Train ↓ | DenseNet | | | | MobileNet | | | | VGG16 | | | |
|---|---|---|---|---|---|---|---|---|---|---|---|---|
| Test → | severity 2 | | severity 3 | | severity 2 | | severity 3 | | severity 2 | | severity 3 | |
| | N (↓) | P (↑) | N (↓) | P (↑) | N (↓) | P (↑) | N (↓) | P (↑) | N (↓) | P (↑) | N (↓) | P (↑) |
| GN-1 | 75.40 | 95.10 | 55.10 | 88.80 | 67.80 | 92.90 | **42.60** | **88.80** | 65.60 | 92.00 | 35.10 | 84.00 |
| UN-1 | 27.10 | 80.80 | 62.40 | 87.30 | 29.70 | 77.70 | 52.80 | 88.30 | **12.40** | **76.10** | 44.60 | 85.20 |
| SPN-1 | **7.90** | **91.00** | **1.00** | **72.70** | **20.50** | **91.40** | 21.80 | 70.40 | **0.00** | **90.90** | **0.00** | **56.80** |
| SN-1 | 73.20 | 95.00 | 51.50 | 93.10 | 62.30 | 90.60 | 41.70 | 87.20 | 71.20 | 91.00 | 44.20 | 87.90 |
| IN-1 | 61.40 | **97.00** | 48.20 | **96.80** | 55.80 | 93.90 | 38.30 | 92.80 | 50.60 | 92.20 | **27.20** | **92.40** |
| SPKN-1 | 86.50 | 96.30 | 61.50 | 94.30 | 81.70 | 93.50 | 56.00 | 90.60 | 84.50 | 92.10 | 63.00 | 89.30 |

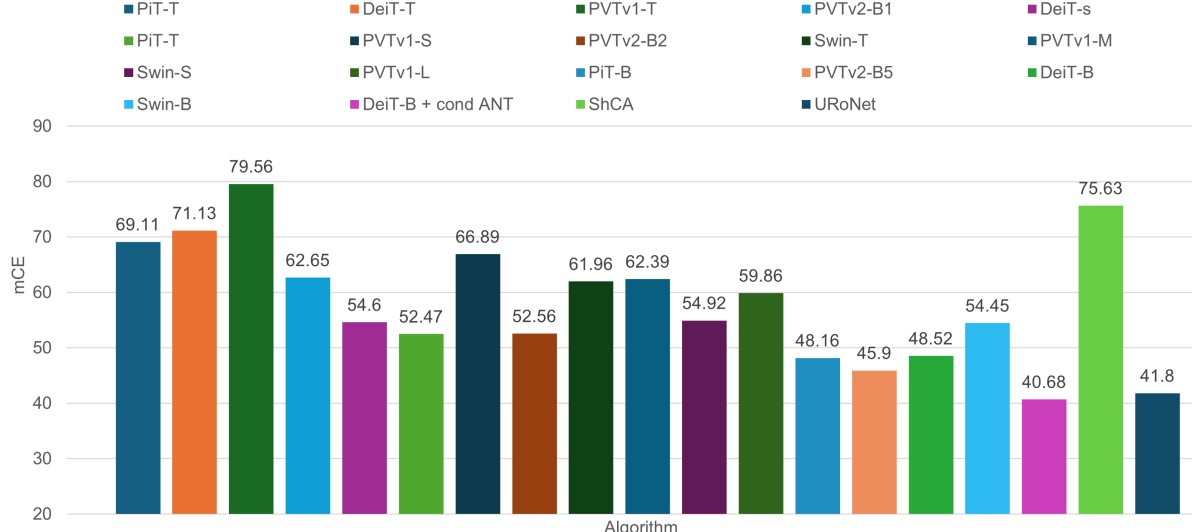

Figure 8: Comparison results against SOTA works on ImageNet-C in terms of mCE (lower is better). The results of the existing algorithms are taken from (Tian et al., 2024), except for ShCA (Zhang et al., 2024).

(ii) somewhat surprisingly, the noise remover not only mitigates corruption but also, certain cases, leads to superior performance compared to clean images. For instance, in case 1, SPN corruption reduces the performance of VGG to **0.00**%, a result that the proposed noise remover restores to **90.90**% and **56.80**% for unseen severity levels two and three, respectively. In case 2, the DenseNet classifier initially achieves 96.10% accuracy on the clean test dataset, which significantly improves to 97.00% when handling Impulse Noise (IN) corruption with severity two, purified using the proposed noise remover trained on IN with severity one.

Similar to the extensive comparison of the CIFAR datasets, an extensive comparison with several vision architectures (Tian et al., 2024) and a recent semi-hard triplet-based data augmentation technique has been performed (Zhang et al., 2024). (Tian et al., 2024) have studied the impact of varying ViT design on yielding robustness against common corruption. Further, based on the robust ViT design, the authors have proposed the defense algorithm utilizing the best ViT model (Deit-B) and a heavy data augmentation based on adversarial noise training (ANT). Whereas, (Zhang et al., 2024) have proposed the semi-hard constraint augmentation (ShCA) technique based on triplet learning where the triplet samples are generated using the online instance distance ranking. The results of this comparison are reported in Figure 8. The proposed algorithm outperforms the majority of the ViT models including the ShCA algorithm except in the case of training the ViT model with heavy data augmentation through ANT. While the difference in mCE value is

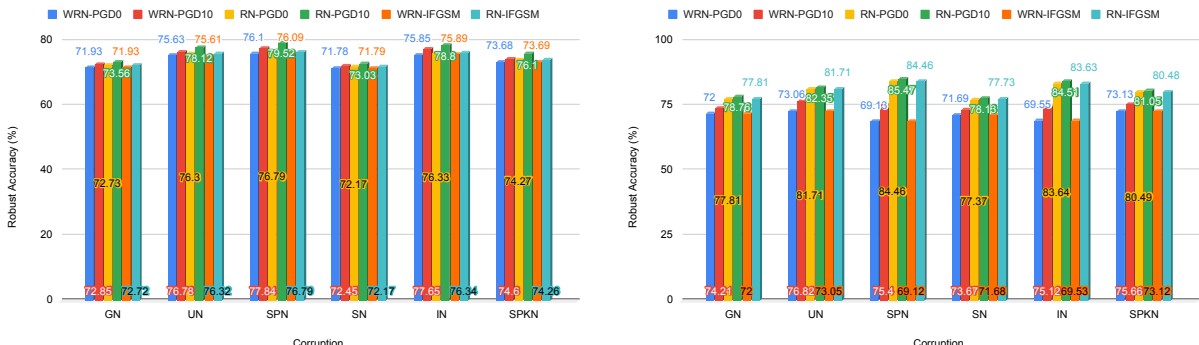

Figure 9: Robust accuracy on adversarial examples on CIFAR-10 where the proposed noise remover is trained on common corruptions. The left figure shows the performance on the ResNet classifier and the right bars show the adversarial mitigation on the WRN16-8 classifier.

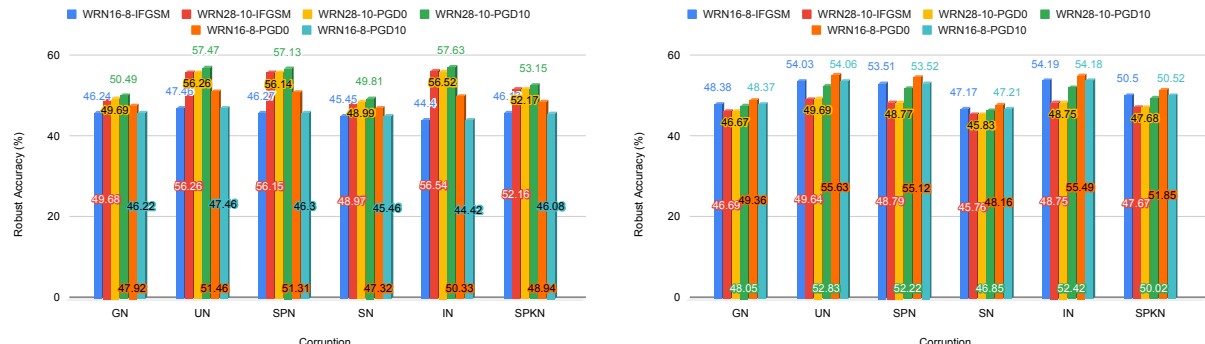

Figure 10: Robust accuracy on adversarial examples on CIFAR-100 where the proposed noise remover is trained on common corruptions. The left figure shows the performance on WRN16-8 and the right bars show the adversarial mitigation on the WRN28-10 classifier.

not drastically different between the proposed URoNet and Deit-B + cond ANT, the computational cost is the biggest advantage of the proposed algorithm.

### 3.2.4 Mitigation of Unseen Adversarial Perturbations

In this section, we study the strength of the proposed noise remover algorithm when trained to remove the common corruption but tested on the adversarial perturbations. The extensive evaluation has been performed using two adversarial perturbation algorithms used above namely PGD and IFGSM. The PGD perturbations are generated for 40 epochs with standard strength $\epsilon = 0.03$ with a random restart value of 0 (PGD0) and 10 (PGD10). The IFGSM perturbation is generated using 40 epochs with strength $\epsilon = 0.03$. Both the perturbations are generated using ResNet and Wide-ResNet16-8 models on the CIFAR-10 dataset and WRN16-8 and WRN28-10 on the CIFAR-100 dataset.

When the classifiers are attacked using the adversarial perturbations, the proposed noise remover which is trained to remove the common corruption shows significant robustness and improves the performance on the adversarial examples. When the ResNet classifier is protected using the GN corrupted trained noise remover, it shows an improvement of 27.41% and 58.7% when the PGD adversarial images are generated from WRN and RN are used, respectively. *Here improvement refers to the relative robust accuracy concerning adversarial accuracy. In other words, the improvement (%) is calculated by subtracting the robust accuracy from the adversarial accuracy. The robust accuracy is the recognition performance on the mitigated images.* The lower improvement in the WRN adversarial examples can be attributed to the fact that the PGD adversarial examples are transferred from WRN to RN and it shows lower attack success. In terms of

Table 9: Comparison of the proposed URoNet with other defense methods in terms of robust accuracy (%).

| Algorithm | CIFAR10 (ResNet) | | CIFAR100 (WRN) | |
|---|---|---|---|---|
| | IFGSM | PGD-40 | IFGSM | PGD-40 |
| N-FGSM (de Jorge Aranda et al., 2022) | 48.10 | 46.70 | 21.30 | 30.80 |
| FACE (Niu & Yang, 2023) | 47.60 | 42.80 | 24.80 | 21.50 |
| AdaIAD (Huang et al., 2023) | 51.47 | 55.01 | 32.31 | 33.76 |
| SD + Reg. (Lukasik et al., 2023) | 48.70 | 39.14 | 14.31 | 9.94 |
| WD + Reg. (Lukasik et al., 2023) | 44.96 | 36.34 | 12.19 | 8.04 |
| **Proposed URoNet** | **76.79** | **76.09** | **48.77** | **52.22** |

corruption, the SPN-trained noise remover shows the best average performance improvement of 42.87%. UN and IN corruption-trained noise removers also performed similarly, behind by 0.4%.

As mentioned above when the adversarial examples are transferred from one CNN to another, they yield lower attack success; and leave less space for improvement. However, when the perturbation is generated from the same network, the attack success is much higher; conversely, the improvement in adversarial mitigation is also higher. For instance, when the PGD0 generated from the WRN is used to attack the WRN classifier, the proposed noise remover shows 44.11% improvement as compared to 5.78% improvement when RN-generated noise is used to attack the classifier. The transference of the adversarial perturbation is an interesting concern where several research works are recently trying to improve the success of the attack against unseen classifiers (Gao et al., 2021; Inkawhich et al., 2019; Li et al., 2020). It is to be noted that the proposed framework does not utilize any knowledge of the perturbation or the classifier and can handle even the highly successful transferable perturbations. The improvement is observed on both the image classifiers and datasets, therefore showing the efficacy of the proposed mitigation algorithm across degradations, datasets, and classifiers. The results on CIFAR-10 and CIFAR-100 datasets are reported in Figure(s) 9 and 10, respectively.

We have further performed the comparison with recent state-of-the-art adversarial mitigation algorithms namely SD + Reg. (Lukasik et al., 2023), WD + Reg. (Lukasik et al., 2023), FACE (Niu & Yang, 2023), N-FGSM (de Jorge Aranda et al., 2022), and AdaIAD (Huang et al., 2023). (Niu & Yang, 2023) compress the low-frequency information and high-frequency information at the spatial level. The defense aims to be agnostic to attack methods and hence an ideal choice for the comparison. (Huang et al., 2023) proposed the adaptive adversarial distillation network which iteratively involves the teacher model in the learning of a student model. (de Jorge Aranda et al., 2022) proposed an adversarial training defense in single-step adversarial attacks without a clip by centering the perturbation concerning noisy samples. As compared to the algorithms mentioned above, the method proposed by (Lukasik et al., 2023) performed the frequency normalization on the convolutional weights. The comparative results shown in Table 9 showcase that the proposed defense which is trained on common noise (SPN noise in this case) surpasses each existing defense by a significant margin in defending unseen adversarial perturbations.

## 3.3 Ablation Studies

The ablation studies are conducted concerning several important components used in the proposed research: (i) varying degradation detection network, (ii) utilization of denoising convolutional autoencoder (DCAE) having a single output, (iii) degradation detection without noise augmentation, and (iv) can we handle grayscale images?

**Degradation Detection Network:** In the proposed research, we have used the Wide-ResNet10-2 (WRN10-2) (Zagoruyko & Komodakis, 2016) for the classification of degradation examples. We have further performed an ablation study using another popular classifier namely VGG (Simonyan & Zisserman, 2014). We have observed across the set of experiments, VGG yields lower performance as compared to WRN10-2. For instance, when the GN corrupted trained detector is tested on the UN-1, the VGG architecture yields 7% lower on the CIFAR-100 dataset as compared to WRN10-2. The performance difference even increased to 22% when the evaluation was performed on the CIFAR-10 dataset.

**Impact of Augmenting Noise:** In the context of the proposed framework, when the noise removed from the images is not combined with the input images, the performance of the degradation detection network experiences a decline of 3% to 6%. This observation underscores the benefits of incorporating auxiliary noise information derived from the proposed noise remover network into the detection network, as opposed to networks relying solely on RGB images. The prime reason for adding the noise can be seen from the fact the noises are external signals to the images whether added intentionally or unintentionally, the absence of knowledge of such an external factor, makes the network generally the network is to generalize against unseen noises. Therefore, the use of noise in the degradation detection network acts as an attack-specific feature to understand that an image might be composed of two signals: clean image and external noise. We assert that such a supervisory signal helps the network better distinguish the clean samples from the noisy images. However, as demonstrated different noises (corruption or adversarial) share an effect through Figure 1 and the impact of frequency components through existing literature (Lukasik et al., 2023; Saikia et al., 2021; Yin et al., 2019b), the inclusion of one type of noise can detect other types of noises.

**DCAE vs. Proposed:** The mitigation performance of the proposed noise remover network with two branches (one as clean image and another as noise pattern) yields at least 6% better performance than the traditional single branch (mapping noisy images to clean images) denoising convolutional autoencoder. The probable reason might be that these traditional denoising systems map each kind of noisy image into ideal clean images without giving importance to the variation present in different noise signals.

**Performance on Gray-scale Images:** To assess the effectiveness of the proposed noise mitigation framework, we conducted experiments using the F-MNIST dataset (Xiao et al., 2017) and developed a custom CNN model consisting of 5 layers. This custom model initially achieved an accuracy of 91.45% on clean test images. The proposed denoiser proved effective in enhancing recognition performance that had been compromised under various types of corruption. For example, in the case of Gaussian Noise (GN), Uniform Noise (UN), and Salt & Pepper Noise (SPN), the performance boost observed amounted to a minimum of 15%, 50%, and 48%, respectively. This analysis demonstrates that the proposed algorithm can accommodate custom models, not limited to wider and deeper networks, and is also capable of enhancing recognition performance, even on grayscale images.

### 3.4 Broader Impact

The broader impact of the presented research can be comprehensively assessed across three key dimensions: (i) classifier, (ii) dataset, and (iii) degradation. These dimensions can be further dissected to highlight the universal applicability of the proposed defense mechanism. For example, within the classifier dimension, we examined various categories, encompassing shallow, deep, and wide architectures. In this study, we evaluated the robustness of each category of classifier/CNN. Shallow custom networks are employed for F-MNIST, VGG, and DenseNet represented deep architectures, while Wide-ResNet epitomized wide architectures. The demonstrated effectiveness of the framework across a spectrum of classifiers affirms that the proposed algorithm can be deployed with any CNN classifier to safeguard against the detrimental effects of corruption.

Within the dataset dimension, we conducted evaluations on multiple aspects, including the number of classes (datasets featuring 10, 100, and 200 classes), image channels (RGB vs. grayscale), and image resolution (ranging from low-resolution CIFAR-10/CIFAR-100 to high-resolution ImageNet). The final dimension we consider is the development of a defense algorithm designed to combat a variety of degradations, spanning common corruptions and adversarial perturbations. These degradations encompass diverse types of corruption, with variations in their severity levels, addressing the critical need for universal resiliency. Our central claim is that the proposed defense mechanism demonstrates a high degree of generalization, which has been rigorously tested across scenarios involving unseen severity levels and varying training-testing configurations related to degradation types. Overall, the research represents a significant stride towards establishing a universal defense system aimed at preserving the robustness and integrity of deep machine learning.

## 4    Conclusion

The integrity of image data, whether subject to common or adversarial forms of corruption, can significantly undermine the performance and practical applicability of neural network-based classification systems. The existing body of literature has traditionally addressed these issues separately, resulting in limited generalizability when confronted with unforeseen forms of corruption (Geirhos et al., 2018; Lopes et al., 2019). In response to this challenge, we have advocated that common corruptions and adversarial perturbations share similarities and should be jointly addressed to create truly resilient algorithms. With this objective in mind, we introduced a unified corruption detection and mitigation framework, which we term as "URoNET". Our findings, based on comprehensive evaluations across multiple publicly available datasets, highlight the effectiveness of this framework in both identifying the presence of corruption in input images and mitigating their detrimental impact to uphold classification performance levels. Additionally, our empirical results suggest a potential connection between common and deliberate corruption, as our network demonstrates scalability across various types of corruption, even successfully addressing intentional corruption while being exclusively trained on common corruption. Future research endeavors in this field could concentrate on further refining the high-level components of the proposed architecture and exploring its application in a wide array of real-world scenarios including the robustness against varying textures and style of images.

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

Table A1: Parameters used for the different severity levels of each common corruption.

| Severity | Gaussian Noise (GN) | Uniform Noise (UN) | Salt & Pepper Noise (SPN) | Shot Noise (SN) | Impulse Noise (IN) | Speckle Noise (SPKN) |
|---|---|---|---|---|---|---|
| 1 | 0.08 | 0.5 | 0.1 | 60 | 0.03 | 0.15 |
| 2 | 0.12 | 0.3 | 0.3 | 25 | 0.06 | 0.2 |
| 3 | 0.18 | 0.1 | 0.5 | 12 | 0.09 | 0.35 |

Table B2: Common corruption detection performance on the CIFAR-10 dataset. The proposed URoNET detection network shows high robustness in handling multiple severities effectively. The lowest severity corruption trained detector generalizes well against seen and unseen corruption. The detailed interpretative caption is given in the text as well.

| Train ↓ | GN | | | UN | | | SPN | | | SN | | | IN | | | SPKN | | |
|---|---|---|---|---|---|---|---|---|---|---|---|---|---|---|---|---|---|---|
| Test → | 1 | 2 | 3 | 1 | 2 | 3 | 1 | 2 | 3 | 1 | 2 | 3 | 1 | 2 | 3 | 1 | 2 | 3 |
| GN-1 | 99.97 | 99.97 | 99.97 | 98.47 | 99.12 | 92.9 | 99.95 | 86.24 | 61.85 | 99.87 | 99.88 | 99.83 | 99.69 | 99.95 | 99.96 | 99.1 | 99.41 | 99.34 |
| GN-2 | 99.98 | 99.98 | 99.98 | 99.79 | 99.59 | 69.00 | 99.98 | 99.98 | 99.98 | 99.92 | 99.97 | 99.98 | 99.98 | 99.98 | 99.98 | 99.4 | 99.8 | 99.95 |
| GN-3 | 87.34 | 99.97 | 100 | 96.93 | 87.48 | 50.00 | 100 | 100 | 100 | 91.95 | 99.79 | 99.94 | 86.46 | 100 | 100 | 66.36 | 90.39 | 99.53 |
| UN-1 | 96.86 | 99.83 | 99.99 | 99.99 | 99.92 | 50.33 | 99.99 | 99.99 | 99.99 | 92.98 | 98.47 | 99.52 | 93.77 | 99.85 | 99.99 | 82.25 | 90.7 | 97.5 |
| UN-2 | 99.98 | 99.98 | 99.98 | 99.98 | 99.98 | 89.09 | 99.98 | 99.98 | 99.98 | 99.94 | 99.96 | 99.97 | 99.97 | 99.98 | 99.98 | 99.59 | 99.88 | 99.94 |
| UN-3 | 99.89 | 99.89 | 99.89 | 98.98 | 99.63 | 99.88 | 99.88 | 99.89 | 99.89 | 99.66 | 99.69 | 99.65 | 61.29 | 99.02 | 99.89 | 98.39 | 98.51 | 98.75 |
| SPN-1 | 96.33 | 99.99 | 100 | 98.35 | 94.28 | 50.0 | 100 | 100 | 100 | 95.49 | 99.84 | 99.95 | 98.75 | 100 | 100 | 77.9 | 94.21 | 99.73 |
| SPN-2 | 52.28 | 93.09 | 99.77 | 88.18 | 54.57 | 50.0 | 99.99 | 100 | 100 | 58.21 | 95.53 | 99.61 | 58.29 | 99.44 | 99.98 | 50.87 | 69.92 | 97.14 |
| SPN-3 | 50.00 | 52.60 | 89.56 | 63.53 | 50.01 | 50.0 | 97.91 | 100 | 100 | 50.08 | 61.33 | 91.25 | 50.01 | 56.53 | 93.64 | 50 | 50.91 | 80.88 |
| SN-1 | 99.99 | 99.99 | 99.99 | 98.86 | 98.55 | 63.67 | 99.99 | 99.99 | 99.99 | 99.99 | 99.99 | 99.99 | 99.05 | 99.99 | 99.99 | 99.99 | 99.99 | 99.99 |
| SN-2 | 99.6 | 99.99 | 99.99 | 97.17 | 95.83 | 50.39 | 99.99 | 99.99 | 99.99 | 99.98 | 99.99 | 99.99 | 98.52 | 99.99 | 99.99 | 99.91 | 99.98 | 99.99 |
| SN-3 | 95.64 | 99.95 | 100 | 98.11 | 91.35 | 50.03 | 100 | 100 | 100 | 99.29 | 99.99 | 100 | 96.81 | 99.99 | 100 | 84.01 | 99.53 | 100 |
| IN-1 | 99.99 | 99.99 | 99.99 | 99.96 | 99.79 | 84.54 | 99.99 | 99.99 | 99.99 | 99.97 | 99.99 | 99.99 | 99.99 | 99.99 | 99.99 | 99.67 | 99.9 | 99.98 |
| IN-2 | 100 | 100 | 100 | 99.91 | 99.63 | 54.91 | 100 | 100 | 100 | 99.88 | 99.98 | 99.99 | 100 | 100 | 100 | 98.54 | 99.7 | 99.95 |
| IN-3 | 98.81 | 99.99 | 100 | 98.01 | 95.57 | 50.02 | 100 | 100 | 100 | 98.84 | 99.9 | 99.97 | 99.73 | 100 | 100 | 88.64 | 97.24 | 99.8 |
| SPKN-1 | 99.96 | 99.96 | 99.96 | 99.04 | 99.04 | 88.93 | 99.96 | 99.96 | 99.96 | 99.96 | 99.96 | 99.96 | 99.96 | 99.96 | 99.96 | 99.96 | 99.96 | 99.96 |
| SPKN-2 | 99.97 | 99.99 | 99.99 | 98.22 | 97.81 | 58.6 | 99.99 | 99.99 | 99.99 | 99.99 | 99.99 | 99.99 | 99.61 | 99.99 | 99.99 | 99.99 | 99.99 | 99.99 |
| SPKN-3 | 98.47 | 99.95 | 99.99 | 97.95 | 94.49 | 50.43 | 99.99 | 99.99 | 99.99 | 99.8 | 99.99 | 99.99 | 99.66 | 99.99 | 99.99 | 99.78 | 99.99 | 99.99 |

# A   Corruption Parameters

Table A1 shows the parameters of different severities used to perform the manipulation of images across corruptions. The prime reason for utilizing different corruption types and multiple severities is that one can not assume the corruption type or its associated severities same across images in the real world. Therefore, the developed defense must be agnostic to not only corruption type but also its severity. Here the severity can be seen as the intensity of the corruption. Figure 4 shows images perturbed by different corruptions with varying severities.

# B   Corruption Detection

The results of common corruption detection on the CIFAR-10 and CIFAR-100 datasets are reported in Table B2 and Table B3, respectively. Each row in the results presents the outcomes when the framework is trained on a specific noise type and severity level, and subsequently tested on images that exhibit various combinations of noise types and severity levels, as denoted in each column. To facilitate a quick assessment of the detection performance, the row corresponding to the lowest detection performance when trained with severity level 1 noise is color-highlighted, providing a clear indication of each corruption's relative strength in the detection process.

When trained using corrupted images generated using UN-3, the detection network yields at least 98.39% detection accuracy except for 61.29% on the testing images with a severity one of the impulse noise (IN-3). It might be due to the low level of severity or imperceptibility of corruption. The phenomenon is observed in each corruption where training with the corruption of low-level severity generalizes well towards the same corruption, unseen corruption, same level of severity, and unseen severity level. Interestingly, the uniform noise with severity three is found complex to be detected by the majority of the noise corruptions. For instance, SPN gives random accuracy on this low level of uniform noise, an approximately random chance accuracy is observed when the detection algorithm is trained on each corruption of shot noise (SN). Moreover, the uniform noise itself when trained with high severity (SV1) is found ineffective in handling the low-level uniform noise. However, a few noises such as GN, IN, and SPKN with the lowest level of severity used in this research showcase their effectiveness in identifying the uniform noise images of

Table B3: Common corruption detection performance on the CIFAR-100 dataset. Common corruption detection performance on the CIFAR-10 dataset. The proposed detection network shows high robustness in handling multiple severities effectively. The lowest severity corruption trained detector generalizes well against seen and unseen corruption.

| Train ↓ | GN | | | UN | | | SPN | | | SN | | | IN | | | SPKN | | |
|---|---|---|---|---|---|---|---|---|---|---|---|---|---|---|---|---|---|---|
| Test → | 1 | 2 | 3 | 1 | 2 | 3 | 1 | 2 | 3 | 1 | 2 | 3 | 1 | 2 | 3 | 1 | 2 | 3 |
| GN-1 | 99.98 | 99.98 | 99.98 | 98.28 | 97.92 | 66.43 | 99.98 | 99.98 | 99.98 | 99.49 | 99.79 | 99.88 | 99.87 | 99.98 | 99.98 | 95.9 | 98.09 | 99.37 |
| GN-2 | 99.94 | 99.99 | 99.99 | 97.18 | 96.01 | 53.28 | 99.99 | 99.99 | 99.99 | 99.63 | 99.89 | 99.95 | 99.51 | 99.99 | 99.99 | 97.56 | 99.14 | 99.73 |
| GN-3 | 98.69 | 99.98 | 99.98 | 99.24 | 97.68 | 50.1 | 99.98 | 99.98 | 99.98 | 95.77 | 99.33 | 99.75 | 98.25 | 99.98 | 99.98 | 85.47 | 93.82 | 98.86 |
| UN-1 | 89.45 | 97.32 | 99.38 | 99.97 | 99.56 | 50.76 | 99.11 | 97.76 | 99.08 | 87.34 | 95.25 | 97.54 | 89.89 | 97.82 | 98.92 | 78.39 | 87.33 | 95.23 |
| UN-2 | 99.97 | 99.99 | 99.98 | 99.99 | 99.99 | 76.67 | 99.4 | 98.14 | 99.8 | 98.91 | 99.37 | 99.33 | 84.67 | 98.15 | 99.23 | 94.47 | 96.45 | 97.51 |
| UN-3 | 99.83 | 99.8 | 98.18 | 96.48 | 99.47 | 99.81 | 74.25 | 69.64 | 52.24 | 99.55 | 99.12 | 93.59 | 55.25 | 69.08 | 73.79 | 98.01 | 97.29 | 87.45 |
| SPN-1 | 81.0 | 97.3 | 99.99 | 91.96 | 78.11 | 50.01 | 100 | 99.98 | 100 | 82.58 | 98.49 | 99.62 | 86.58 | 99.98 | 100 | 66.9 | 86.13 | 98.05 |
| SPN-2 | 70.4 | 97.07 | 99.93 | 95.81 | 80.92 | 50.01 | 99.98 | 99.8 | 99.98 | 78.26 | 97.32 | 99.66 | 86.44 | 99.85 | 99.98 | 67.19 | 84.82 | 98.23 |
| SPN-3 | 50.02 | 55.31 | 90.29 | 68.26 | 50.01 | 50 | 98.07 | 62.24 | 100 | 50.04 | 66.05 | 91.68 | 50.03 | 62.05 | 95.35 | 50.01 | 50.82 | 82.50 |
| SN-1 | 99.98 | 99.98 | 99.98 | 95.67 | 96.63 | 75.84 | 99.91 | 96.79 | 99.98 | 99.98 | 99.98 | 99.98 | 64.7 | 96.72 | 99.69 | 99.97 | 99.98 | 99.84 |
| SN-2 | 99.95 | 99.95 | 99.95 | 99.33 | 98.59 | 77.6 | 99.95 | 99.95 | 99.95 | 99.95 | 99.95 | 99.95 | 99.94 | 99.95 | 99.95 | 99.94 | 99.95 | 99.95 |
| SN-3 | 99.51 | 99.98 | 99.98 | 98 | 95.53 | 51.11 | 99.98 | 99.98 | 99.98 | 99.98 | 99.98 | 99.98 | 99.92 | 99.98 | 99.98 | 99.35 | 99.95 | 99.98 |
| IN-1 | 99.96 | 99.99 | 99.99 | 99.51 | 98.26 | 51.79 | 99.99 | 99.99 | 99.99 | 99.3 | 99.92 | 99.98 | 99.99 | 99.99 | 99.99 | 96.1 | 98.54 | 99.80 |
| IN-2 | 99.54 | 99.98 | 99.98 | 99.12 | 96.85 | 50.06 | 99.98 | 99.98 | 99.98 | 97.9 | 99.73 | 99.96 | 99.98 | 99.98 | 99.98 | 90.68 | 97.03 | 99.60 |
| IN-3 | 75.58 | 98.62 | 100 | 94.06 | 82.29 | 50.01 | 100 | 100 | 100 | 77.58 | 96.93 | 99.33 | 86.86 | 100 | 100 | 65.14 | 79.45 | 99.96 |
| SPKN-1 | 99.93 | 99.97 | 99.97 | 93.11 | 94.51 | 68.8 | 99.85 | 96.47 | 99.97 | 99.97 | 99.97 | 99.97 | 67.93 | 96.49 | 99.7 | 99.97 | 99.97 | 99.97 |
| SPKN-2 | 99.9 | 99.9 | 99.9 | 99.55 | 99.26 | 93.17 | 99.9 | 99.9 | 99.9 | 99.9 | 99.9 | 99.9 | 99.9 | 99.9 | 99.9 | 99.9 | 99.9 | 99.90 |
| SPKN-3 | 97.67 | 99.98 | 99.98 | 96.52 | 94.02 | 50.56 | 99.98 | 99.98 | 99.98 | 99.98 | 99.98 | 99.98 | 99.89 | 99.98 | 99.98 | 99.92 | 99.98 | 99.98 |

Table B4: Adversarial example detection on the CIFAR-10 and CIFAR-100 datasets when the detectors are trained on each common corruption of varying severity. Colored boxes highlight the best performing common corruptions in identifying adversarial examples.

| Train → | GN | | | UN | | | SPN | | | SN | | | IN | | | SPKN | | |
|---|---|---|---|---|---|---|---|---|---|---|---|---|---|---|---|---|---|---|
| Test ↓ | 1 | 2 | 3 | 1 | 2 | 3 | 1 | 2 | 3 | 1 | 2 | 3 | 1 | 2 | 3 | 1 | 2 | 3 |
| Evaluation on the CIFAR-10 dataset | | | | | | | | | | | | | | | | | | |
| IFGSM | 50.06 | 50.01 | 50.00 | 50.01 | 50.07 | 82.86 | 50.00 | 50.00 | 50.00 | 50.05 | 50.01 | 50.00 | 50.05 | 50.01 | 50.0 | 50.81 | 50.05 | 50.00 |
| PGD | 50.43 | 50.02 | 49.97 | 49.97 | 50.24 | 99.46 | 49.98 | 49.97 | 50.00 | 50.42 | 49.94 | 49.97 | 50.08 | 49.97 | 49.96 | 58.24 | 50.41 | 49.96 |
| Evaluation on the CIFAR-100 dataset | | | | | | | | | | | | | | | | | | |
| IFGSM | 50.02 | 50.00 | 50.00 | 50.08 | 50.04 | 87.23 | 50.00 | 50.0 | 50.0 | 50.16 | 50.1 | 50.02 | 50.0- | 50.01 | 50.01 | 50.12 | 50.91 | 50.02 |
| PGD | 50.15 | 50.00 | 50.01 | 50.12 | 50.12 | 99.49 | 50.01 | 50.00 | 50.00 | 52.6 | 51.95 | 50.14 | 50.02 | 50.01 | 50.01 | 51.53 | 61.04 | 50.02 |

low severity. The SPKN with a severity one even surpasses the lowest detection accuracy of UN-3 and yields at least 88.93% detection accuracy.

Similar behavior can be observed on the CIFAR-100 dataset where the SPKN-1 shows the highest level of performance by achieving the highest detection accuracy across the range of corruptions. Although the problem is still a two-class classification; however, the corruption detection performance suffers significantly when moved from CIFAR-10 to the CIFAR-100 dataset. For example, on the CIFAR-10 dataset, the lowest performance achieved by SPKN-1 is 88.93% as compared to the lowest performance (67.93%) on the CIFAR-100 dataset. In contrast, the Gaussian noise with severity one which shows the lower generalizability against the SPN with severity 3 (61.85%) on the CIFAR-10 dataset does not show any vulnerability in handling the SPN noise on the CIFAR-100 dataset. Overall, Gaussian noise and SPKN noise show a higher level of efficiency in the detection of common corruption which is extensively tested in several generalized settings.

# C    Corruption Mitigation

Table C6 and C7 show the mitigation results on the CIFAR-10 and CIFAR-100 datasets, respectively. *For both these tables, the numbers in row iN represent the performance of the CNN when under the effect of the common corruption generated with $i^{th}$ severity level. The row with iP contains the mitigation performance of the $i^{th}$ severity corruption using the proposed algorithm. For instance, GN-1N: accuracy on samples corrupted with severity-1 Gaussian noise. GN-1P-1 represents the mitigation accuracy on severity-1 Gaussian noise corrupted examples when the proposed noise remover is trained using severity-1 examples.* In numerical terms: GN-1N can reduce the accuracy of Wide-ResNet28-10 classifier 24.71%, which increases to 51.95% (GN-1P-1), 47.31% (GN-1P-2), 36.85% (GN-1P-3) when the proposed noise remover is trained using the Gaussian noise images with severity 1, 2, and 3, respectively.

Table B5: Common corruption detection on the CIFAR-10 and CIFAR-100 datasets where the detectors are trained on adversarial perturbations. In contrast to adversarial examples detection trained on common corruption, adversarial detector shows significant success in identifying common corruption. Colored boxes highlight the strongest common corruption (yielding the lowest accuracy) to be defended by the detectors.

| Test → | GN | | | UN | | | SPN | | | SN | | | IN | | | SPKN | | |
|---|---|---|---|---|---|---|---|---|---|---|---|---|---|---|---|---|---|---|
| Train ↓ | 1 | 2 | 3 | 1 | 2 | 3 | 1 | 2 | 3 | 1 | 2 | 3 | 1 | 2 | 3 | 1 | 2 | 3 |
| Evaluation on the CIFAR-10 dataset | | | | | | | | | | | | | | | | | | |
| FGSM40 | 97.65 | 97.65 | 97.65 | 96.86 | 97.11 | 97.43 | 97.65 | 97.65 | 97.65 | 97.65 | 97.3 | 97.62 | 87.02 | 97.5 | 97.65 | 97.5 | 97.47 | 97.5 |
| PGD40 | 99.87 | 99.73 | 99.42 | 95.06 | 97.96 | 99.26 | 92.01 | 97.41 | 97.83 | 99.52 | 98.76 | 97.69 | 54.45 | 79.04 | 90.31 | 97.02 | 95.66 | 90.71 |
| Evaluation on the CIFAR-100 dataset | | | | | | | | | | | | | | | | | | |
| IFGSM40 | 99.61 | 99.6 | 99.61 | 94.91 | 95.52 | 97.25 | 99.55 | 95.96 | 99.61 | 99.45 | 99.45 | 99.39 | 60.33 | 95.99 | 99.44 | 98.54 | 98.58 | 98.7 |
| PGD20 | 99.6 | 99.6 | 99.6 | 93.27 | 94.88 | 96.56 | 98.37 | 86.13 | 99.6 | 99.42 | 99.3 | 99.19 | 53.55 | 86.3 | 97.2 | 98.26 | 98.11 | 97.42 |

Table C6: Common corruption mitigation on the CIFAR-10 dataset using ResNet and Wide-ResNet16-8. In brief, the noise remover trained with severity coming at the test time shows better robustness as compared to training on different severity levels. However, the unseen severity training can boost the performance multiple folds. The detailed interpretative caption is given in the text as well.

| Data | Mitigation Severity | | | | | | | | | | | | | | | | | |
|---|---|---|---|---|---|---|---|---|---|---|---|---|---|---|---|---|---|---|
| | GN | | | UN | | | SPN | | | SN | | | IN | | | SPKN | | |
| | 1 | 2 | 3 | 1 | 2 | 3 | 1 | 2 | 3 | 1 | 2 | 3 | 1 | 2 | 3 | 1 | 2 | 3 |
| ResNet: 91.81% clean accuracy | | | | | | | | | | | | | | | | | | |
| 1N | | 32.97 | | | 13.84 | | | 20.48 | | | 32.73 | | | 51.73 | | | 44.37 | |
| 1P | 75.97 | 71.92 | 64.04 | 28.24 | 31.22 | 40.72 | 83.62 | 79.05 | 73.67 | 76.62 | 70.06 | 59.82 | 82.9 | 83.28 | 78.1 | 78.95 | 76 | 68.98 |
| 2N | | 19.46 | | | 15.4 | | | 13.14 | | | 18.13 | | | 30.82 | | | 32.51 | |
| 2P | 71.69 | 72.13 | 64.68 | 42.45 | 48.15 | 60.0 | 81.4 | 78.79 | 73.33 | 71.08 | 69.72 | 62.02 | 82.9 | 82.95 | 78.35 | 77.0 | 75.92 | 68.62 |
| 3N | | 15.22 | | | 18.19 | | | 10.96 | | | 14.97 | | | 21.52 | | | 18.6 | |
| 3P | 56.27 | 63.68 | 63.34 | 58.12 | 65.24 | 69.47 | 64.79 | 75.03 | 71.92 | 57.12 | 63.04 | 62.75 | 82.46 | 83.14 | 77.93 | 61.0 | 66.53 | 68.55 |
| Wide-ResNet16-8: 93.41% clean accuracy | | | | | | | | | | | | | | | | | | |
| 1N | | 45.27 | | | 15.97 | | | 20.2 | | | 45.33 | | | 63.52 | | | 58.17 | |
| 1P | 80.05 | 75.81 | 66.76 | 30.83 | 34.42 | 42.94 | 87.07 | 82.45 | 77.01 | 80.44 | 73.97 | 62.37 | 86.25 | 86.34 | 82.37 | 82.54 | 80.15 | 72.34 |
| 2N | | 25.06 | | | 16.55 | | | 11.09 | | | 25.72 | | | 35.79 | | | 46.02 | |
| 2P | 75.74 | 75.8 | 67.36 | 45.45 | 51.96 | 62.97 | 85.27 | 82.06 | 76.85 | 75.15 | 74.23 | 65.32 | 86.33 | 86.43 | 82.6 | 80.59 | 79.4 | 72.7 |
| 3N | | 15.16 | | | 21.72 | | | 10.32 | | | 17.77 | | | 21.74 | | | 26.0 | |
| 3P | 60.84 | 67.61 | 67.76 | 61.28 | 69.88 | 73.67 | 69.99 | 79.14 | 75.26 | 61.8 | 67.51 | 66.03 | 86.21 | 86.01 | 81.84 | 64.81 | 70.26 | 71.79 |

To extensively study the impact of noise removal, we have performed the experiments under both the same and different severity levels. For instance, when the proposed noise remover is trained using severity-1 of Gaussian noise, it can increase the performance to 75.97%, 71.69%, and 56.27% when the corrupted images with severity one (32.97%), two (19.46%), and three (15.22%) used for classification, respectively. This showcases that the proposed algorithm can mitigate corruption regardless of severity differences. In most cases of severe common corruptions, the performance boost is as high as 6 times (10.96% to 64.79 in case of RN) to 7 times (10.32% to 69.99% in case of WRN).

Similar to the CIFAR-10 dataset, the experiments are extensively performed under both the same and different corruption severity. The experiments are performed using two wider and deeper architectures namely Wide-ResNet28-10 (WRN28-10) and Wide-ResNet16-8 (WRN16-8). The analysis can be broken down broadly into two pieces: (i) degradation impact on the performance of both classifiers under different common corruption and (ii) improvement in the recognition accuracy when the images are purified using the proposed noise remover algorithm.

The WRN28-10 and WRN16-8 yield an accuracy of 76.21% and 74.57% on the clean images of the CIFAR-100 dataset, respectively. However, the performance suffers hugely when the distribution of the testing images changes due to different corruptions. As observed earlier, the WRN16-8 architecture being shallow as compared to WRN28-10 is found less robust in handling corruption. For instance, the performance of the WRN28-10 classifier drops to 22.95%, and the performance of WRN16-8 drops further to 17.44% when the shot noise (SN) with severity one is used to attack the classifiers. The impact of other common corruption also gets noticed significantly on both the classifiers and hence we reaffirm that no corruption should be ignored and the severity of the corruption is also an important factor to consider.

Table C7: Common corruption mitigation on the CIFAR-100 dataset using Wide-ResNet28-10 and Wide-ResNet16-8. Common corruption mitigation on the CIFAR-10 dataset using ResNet and Wide-ResNet16-8. In brief, the noise remover trained with severity coming at the test time shows better robustness as compared to training on different severity levels. However, the unseen severity training can boost the performance multiple folds. The detailed interpretative caption is given in the text as well.

| Data | Mitigation Severity | | | | | | | | | | | | | | | | | |
|---|---|---|---|---|---|---|---|---|---|---|---|---|---|---|---|---|---|---|
| | GN | | | UN | | | SPN | | | SN | | | IN | | | SPKN | | |
| | 1 | 2 | 3 | 1 | 2 | 3 | 1 | 2 | 3 | 1 | 2 | 3 | 1 | 2 | 3 | 1 | 2 | 3 |
| Wide-ResNet28-10: 76.21% clean accuracy | | | | | | | | | | | | | | | | | | |
| 1N | | 24.71 | | | 2.51 | | | 6.64 | | | 22.95 | | | 26.81 | | | 31.6 | |
| 1P | 51.95 | 47.31 | 36.85 | 17.68 | 12.62 | 9.22 | 60.92 | 54.52 | 45.39 | 50.89 | 45.32 | 36.11 | 61.82 | 61.43 | 60.41 | 54.7 | 52.1 | 41.71 |
| 2N | | 12.5 | | | 4.35 | | | 10.88 | | | 10.87 | | | 10.79 | | | 20.82 | |
| 2P | 46.6 | 46.41 | 37.74 | 34.62 | 24.54 | 18.44 | 61.0 | 54.72 | 45.05 | 43.71 | 45.0 | 36.68 | 61.64 | 61.34 | 60.1 | 50.99 | 51.55 | 42.62 |
| 3N | | 6.57 | | | 9.44 | | | 1.00 | | | 6.67 | | | 6.99 | | | 8.18 | |
| 3P | 31.55 | 36.75 | 37.35 | 43.78 | 40.07 | 31.23 | 39.13 | 48.46 | 45.99 | 29.51 | 35.52 | 37.44 | 61.1 | 61.1 | 60.25 | 31.13 | 39.64 | 42.56 |
| Wide-ResNet16-8: 74.57% clean accuracy | | | | | | | | | | | | | | | | | | |
| 1N | | 20.09 | | | 2.26 | | | 4.12 | | | 17.44 | | | 21.59 | | | 25.68 | |
| 1P | 53.17 | 48.46 | 38.55 | 18.75 | 12.68 | 8.49 | 61.18 | 55.69 | 46.5 | 52.32 | 46.2 | 37.42 | 62.38 | 61.5 | 60.77 | 55.86 | 53.29 | 43.27 |
| 2N | | 9.17 | | | 3.37 | | | 8.02 | | | 7.78 | | | 7.97 | | | 15.4 | |
| 2P | 48.23 | 47.88 | 39.31 | 35.87 | 25.06 | 18.24 | 61.06 | 55.58 | 46.07 | 45.5 | 46.57 | 38.48 | 62.19 | 61.55 | 60.92 | 52.15 | 52.53 | 44.02 |
| 3N | | 4.03 | | | 6.7 | | | 1.06 | | | 3.88 | | | 4.66 | | | 6.27 | |
| 3P | 33.44 | 38.11 | 39.37 | 45.35 | 42.12 | 32.13 | 41.27 | 49.76 | 47.18 | 30.12 | 38.18 | 38.87 | 61.7 | 61.47 | 60.7 | 31.71 | 40.91 | 43.42 |

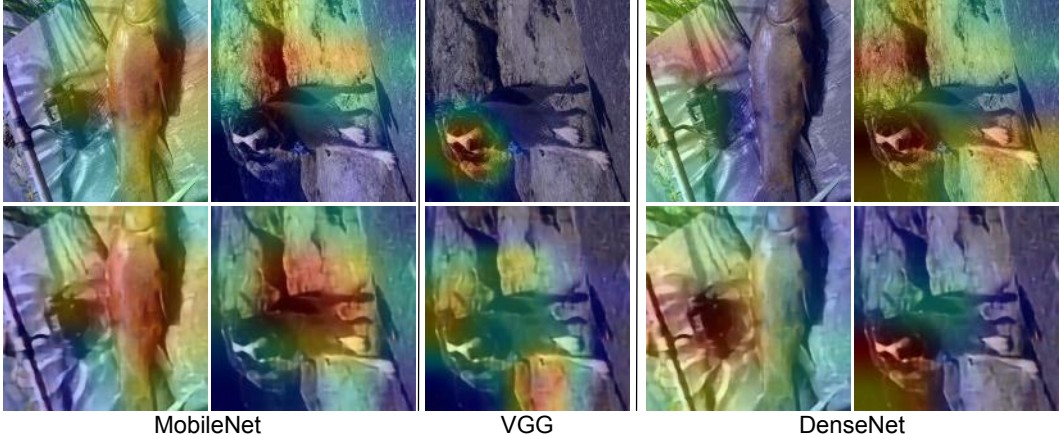

MobileNet       VGG       DenseNet

Figure D1: Grad-CAM (Selvaraju et al., 2017) visualization on the clean and purified images shows that the proposed framework can restore the focus of the CNNs to the critical regions. The first row shows the visualizations of the clean images and the last row shows the purified images.

## D  Grad-CAM Visualization

As shown in Figure D1, we can discern that the proposed framework effectively guides various classification networks to concentrate on more classification-relevant localized regions within the purified images. Interestingly, in some cases, this focus appears to be even superior to the attention exhibited by the CNNs when processing real, clean images. This phenomenon could be attributed to the improved recognition performance, which would otherwise be compromised by the presence of image corruption and the network's consideration of corresponding focus regions.

