# OpenReview forum: "Treating Adversarial and Natural Noise Equally: Building a Robust Network for Unified Resilience"
_TMLR — Rejected by TMLR_

### Review · Reviewer_Sb6p · 2024-12-10

**Summary Of Contributions:**

This manuscript argues that common image corruptions and adversarial perturbations can be addressed in a unified framework for both detection and removing. The proposed pipeline, named URoNet, performs three tasks:
 1. image perturbation removal;
 2. binary image perturbation classification.
 3. image classification, which consumes the images removed perturbations by (1).

**Audience:**

No

**Claims And Evidence:**

No

**Requested Changes:**

1. I would like to suggest the authors put efforts to improve the clarity and flow of writing by adhering to academic conventions.
2. I would like to advise the authors to re-state the motivation. The motivation that the perturbations imperceptible to human eyes is not sufficient.
2. Ensure consistency and adherence to standard notations within the deep learning community.
3. Adjust the writing of the experiments and significantly shorten the writing. Condense non-critical information and streamline the presentation of experiments for improved readability.

**Strengths And Weaknesses:**

# Strengths

The pipeline proposed in this manuscript may have practical applications in certain scenarios.

# Weaknesses

 1. Limited research contribution. While the manuscript presents a potentially useful pipeline, it lacks sufficient novel insights or theoretical contributions to stand out as a significant research work.
2. Poor presentation and not following convention in academic writing. The manuscript is dense and poorly written, requiring significant effort to read and understand. The English does not conform to academic writing standards. For instance:
    - In the second paragraph, “it’s arduous” should be “it is arduous.”
    - Equations are not properly integrated into the text. For example, in Section 2.1, Equation (1) lacks a full stop. The subsequent sentence beginning with “Here…” should start with “where…” in lowercase.
    - Notations. For example, in section 2.1, the fifth paragraph, the notation $M\*N\*c$ is unusual.
3. Possible AI-generated content. Certain sections of the text appear generated by AI, contributing to inconsistencies and lack of clarity.
4. Weak Motivation.
    - The manuscript’s primary motivation—that perturbations appear similar to the human eye—is insufficient justification for unifying the pipeline into perturbation removal and detection.
    - The discussion of this point spans three lengthy paragraphs, making it overly dense and reducing readability. Revising and condensing the motivation section is strongly advised.
5. Inconsistent Notations.
    - The notations lack consistency and do not align with conventions in the deep learning community. For instance:
	- It is unclear what $F(x,y)$ represents in Section 2.1.
	- The choice of $f(x,y)$ to denote an image and $\eta(x,y)$ for noise is non-standard in deep learning community and potentially confusing.
6. Unnecessary length/dense.
     - The manuscript is overly lengthy and resembles an experimental report rather than a concise research paper.
     - Certain sections, such as Table 1, could be moved to the appendix without affecting the main narrative flow.
7.  Poor Experiment Presentation.
     - The experimental section lacks clear organization and is difficult to follow. The results are not presented in a reader-friendly mean, reducing their impact and interpretability.

---

> ### Author Response · Authors · 2024-12-29
> **Response**
>
> **Editorial Changes:** We want to thank the reviewers for their constructive comments. Based on the suggestions of the reviewer, we have again revisited the notations and changed them accordingly to fit the current deep learning literature. Further, we have carefully proofread the paper and updated it to avoid any errors. We want to highlight that there is no AI content in the paper.
>
> For example, based on the suggestion of the reviewer, we have carefully do the adjustment of the materials including the movement of Table 1 to the appendix. Further, we have condensed Tables 1 and 2 provided the average detection accuracy in the main paper, and moved the detailed Tables to the appendix (Tables B2 and B3). Similarly, Tables 7 and 8 (now Tables C6 and C7) are compressed to the key results and replaced with Figures 6 and 7.
>
>
> **Notation:** $f$ represents the input clean image, $\eta$ represents the noise vector/image, which can be any (common corruption or an adversarial perturbation). The modification of $f$ through $\eta$ results in an attacked image “$g$”. Since the noise vector extracted from the noise remover network (part “A” of Figure 3) is of the same dimension as the input; therefore, after its concatenation with an image along the channel ($c$) dimension, the number of channels will be doubled which is mentioned through description and notation of $M\times{N}\times{2c}$.
>
> **Motivation:** We have rewritten the introduction section to replace the over-emphasis on the perceptibility of the attacks on human examiners with the effect of corruption and adversarial attacks on the frequency components and a possible connection explored between corruption and adversarial perturbation explored in the literature. It is observed that the corruptions and perturbations leave a similar impact on low and high-frequency components. The possible correlation between corruption and perturbation has been recently explored as well for a unified detection network [B,C]. Moreover, these preliminary studies have used the ImageNet classifiers for detecting the corruptions and adversarial perturbations without using any attack knowledge and hence not found generalized in handling unseen attacks. However, such preliminary studies motivate this study to develop a unified, robust, and generalized architecture that can effectively handle corruption and perturbation. Further, the proposed architecture must be agnostic to the target classifier so that it can be easily applied to any classifier. The existing defenses whether regularization-based [A] or frequency-based [D,E], are highly sensitive to the configuration of a target classifier as mentioned above.
>
> *We want to highlight that this is the first work that is handling common corruption and adversarial attacks comprehensively. We have proposed a unique URoNet architecture that aims to remove the noise and detect the noisy images with high accuracy and is found generalized in handling unseen adversarial perturbations.*
>
> **Comparison:** Further, the recent works highlighted by the reviewer ijzf show limited robustness. For example, the SD + Reg [6] obtained a robust accuracy of 56.89% when the ResNet-9 model was used; however, the accuracy drastically dropped to 44.42% when a deeper version (ResNet-20) was used. Since the proposed UroNet does not utilize any knowledge of a target network or its size (it is a separate autoencoder network), is not sensitive to these changes.
>
> We have extended the comparison with frequency-based approaches as suggested by reviewer ijzf and found that the frequency-based approaches, including [A], are sensitive to the architecture design. On top of that, the approach in [A] failed drastically when the attack was applied in the frequency domain. Whereas, the proposed approach does not utilize any knowledge of adversarial or target architecture.
>
> **An extensive comparison with several other state-of-the-art approaches can be found in Tables 6 to 9. The proposed URoNet surpasses the existing approaches by a significant margin, demonstrating its effectiveness in handling seen or unseen complex attacks.**
>
> [A] Jovita Lukasik, Paul Gavrikov, Janis Keuper, and Margret Keuper. Improving native CNN robustness with filter frequency regularization. TMLR 2023.
>
> [B] Akshay Agarwal, Nalini Ratha, Mayank Vatsa, and Richa Singh. Benchmarking robustness beyond lp norm adversaries. In European Conference on Computer Vision, pp. 342–359. Springer, 2022.
>
> [C] Akshay Agarwal, Nalini Ratha, Mayank Vatsa, and Richa Singh. Exploring robustness connection between artificial and natural adversarial examples. In IEEE/CVF Conference on Computer Vision and Pattern Recognition, pp. 179–186, 2022.
>
> [D] Liao KH, Yeh CY, Chen HW, Chen MS. Evaluating Adversarial Robustness in the Spatial Frequency Domain. arXiv preprint arXiv:2405.06345. 2024.
>
> [E] Tonmoy Saikia, Cordelia Schmid, and Thomas Brox. Improving robustness against common corruptions with frequency-biased models. ICCV 2021

---

> ### Comment · Reviewer_Sb6p · 2025-01-22
> **Thanks for the replies.**
>
> Dear authors,
>
> Thanks for the efforts of addressing my concerns.
> I appreciate it.
>
> Reviewer

---

### Review · Reviewer_ijzf · 2024-12-19

**Summary Of Contributions:**

This paper presents a unified framework for detecting and mitigating various types of degradation, ranging from common corruptions to adversarial perturbations, called URoNet. This approach facilitates noise removal from images through a dedicated removal network, which also enables the detection of noisy images. After generating a clean image, the final component of this framework is an object detection network. The proposed network is trained on diverse types of corruption with varying severity levels and is evaluated on both seen and unseen types of degradation. The backbone architectures employed in this approach include different convolutional networks, such as ResNet and WideResNet16-8. The method is evaluated on several datasets, including CIFAR-10, CIFAR-100, and TinyImageNet.

**Audience:**

Yes

**Claims And Evidence:**

Yes

**Requested Changes:**

1. This paper would benefit from a detailed discussion on why degenerations should be included in the training process (which is the idea of adversarial training), as well as its impact on the accuracy of object classification tasks when applied to clean inputs in this framework and as a sole classification task.

2. It is advisable to include an evaluation on various other robustness datasets (see Weaknesses).

3. A discussion comparing frequency-based approaches for improving robustness with the proposed method would also be beneficial. A comparison of the proposed method with one of the mentioned would be sufficient.

4. Last, a comparison or usage of robust sota networks would be helpful.

**Strengths And Weaknesses:**

**Strengths**
- This paper tackles an important topic of the improvement of network’s robustness through unified detection and mitigation framework.
- The framework has shown good results. Extensive experiments demonstrate improved results across different severity levels and types of degradation, both for seen and unseen types.

**Weaknesses**
- Equation 2 is unclear to me: What is the input to the degradation network? During training the input to the degeneration network is a concatenation of the noise patterns ($\eta'_f(x,y)$ and $\eta'_g(x,y)$), which are zeros for a clean image. However during inference the input contains the clean image and the reconstructed noise ($h'(x,y)$ and $\eta'_h(x,y)$). So this is not clear to me.
- The actual results are difficult to interpret. It would be better to use accuracy instead of improvements to allow for a clearer comparison of the steps (see Fig. 6 and Fig. 7).
- In general, it is hard to assess the actual improvements due to the lack of clear readability of the tables.
- Only older CNN architectures are being used. It would be better to use rather currently common used ones, to see the benefit of the proposed method (e.g., ConvNext, transformer models). Furthermore, this method *only* investigates CNNs.
- How to the computation costs compare with other methods?
- This paper would benefit from evaluations of other types of *robustness* datasets to actually show the ability of this method to detect the noise and classify the object in one step. Possibilities would include e.g., ImageNet-A/R and the cue-conflict dataset [8].
- In Figure 5, the Grad-CAM visualization does not appear to show a clear focus on the clean image, which results in the question about the benefit of this part of the analysis in general.
- Though this paper shows good results on FGSM and PGD, it would be interesting to see, how this method behaved on stronger more difficult attacks, such as AutoAttack [9]
- What is the accuracy of the backbone only trained for object recognition and the accuracy within the proposed URoNET framework? A comparison would be helpful.

*General issue:*
 The framework is based on the fact that the noise needs to be present during training. Overall, there is a lack of explanation about the rationale. Currently, the approach seems to mirror the idea of adversarial training by including common corruptions into the training process, albeit without the reduced accuracy typically associated with adversarial training. In training, the corruptions or noise must be known beforehand to train the corruption detection network. I am not sure this approach is beneficial against methods, which try to improve the robustness of networks without the inclusion of degradations at test time (see my discussion below).  How much improvement does this method achieve compared to stronger and different types of degradations at test time (e.g. AutoAttack, ImageNet-R)? This would showcase the benefit of the network.


*General Comment:*
There is a missing discussion on frequency analysis. Several studies have analyzed the performance of image classification models based on their frequency characteristics. For instance, Wang et al. revealed that image classification models often perform well on high-pass filtered images, which appear noisy and lack discriminative information, while low-pass filtered images, which are easily recognizable to humans, are frequently misclassified. This spectral bias toward high-frequency bands is associated with both types of degradation mentioned in this paper—adversarial robustness and common corruptions—since both primarily affect high-frequency bands. Most work attempts to avoid using adversarial training or to incorporate any kind of degradation in the training process to enhance robustness by increasing the low-frequency bias in the neural network ([1 -7]). A small comparison to this kind of work would show the improvements by including common corruptions into the training process.


*Minor Comment:* In Figures 1 and 2, I would recommend to use the same subheadings for both figures for easier readability and comparability.


[1]Paul Gavrikov and Janis Keuper. Can Biases in ImageNet Models Explain Generalization? CVPR, 2024

[2] Haohan Wang, Xindi Wu, Zeyi Huang, and Eric P. Xing.High-frequency component helps explain the generalization of convolutional neural networks. CVPR 2020

[3] Antonio A. Abello, Roberto Hirata, and Zhangyang Wang.  Dissecting the high-frequency bias in convolutional neural networks. CVPR Workshops 2021

[4] Dong Yin, Raphael Gontijo Lopes, Jon Shlens, Ekin Dogus Cubuk, and Justin Gilmer. A fourier perspective on model robustness in computer vision. NeurIPS 2019

[5] Raphael Gontijo Lopes, Dong Yin, Ben Poole, Justin Gilmer, and Ekin D. Cubuk. Improving robustness without sacrificing accuracy with patch gaussian augmentation, 2020

[6] Jovita Lukasik, Paul Gavrikov, Janis Keuper, and Margret Keuper. Improving native CNN robustness with filter frequency regularization. TMLR 2023.

[7] Tonmoy Saikia, Cordelia Schmid, and Thomas Brox. Improving robustness against common corruptions with frequency biased models. ICCV 2021

[8] Robert Geirhos, Patricia Rubisch, Claudio Michaelis, Matthias Bethge, Felix A. Wichmann, and Wieland Brendel. Imagenet-trained cnns are biased towards texture; increasing shape bias improves accuracy and  robustness. ICLR 2019

[9] Francesco Croce and Matthias Hein. Reliable evaluation of adversarial robustness with an ensemble of diverse parameter-free attacks. ICML 2020

---

> ### Author Response · Authors · 2024-12-29
> **Response to Requested Changes**
>
> **Response to General Issue:** We want to thank the reviewers for pointing out a very important and critical property of any defense network, i.e., generalizability. While we agree that adversarial training and the proposed URoNet utilize noise during the training of the defense network; however, there are several differences between the approaches: While adversarial training augments the clean data with adversarial images to improve the robustness, we aim to remove the noise and use the noise as an auxiliary signal along with clean images for training the detection network. In other words, the proposed approach does not have the two images but a clean image and a possible common corruption in it. Further, adversarial training is a computationally heavy process, whereas, the proposed approach is a lightweight process and is agnostic to the target classifier.
>
> **Generalization of URoNet:** In response to the use of noise at the time of training and the practicality of the proposed approach, we have performed extensive experiments under several generalized settings: (i) unseen noise setting (where the adversarial noise and common corruption at the time of evaluation can be completely unseen. It is to be noted that, due to our assumption that adversarial noise and common corruption have similar attack attributes, we have never used adversarial noises in the training of the URoNet as mentioned in section 3.1.3 and section 3.2.4). For example, the results shown in Tables 1 and 2 are reported under unseen corruption settings and Figures 8 and 9 under unseen adversarial noise mitigation settings. (ii) unseen model: where the training and target model are unseen. For example, URoNet can be seen as a plug-in network that does not have anything to do with any target network and hence can be easily used to protect any network.  (iii) unseen dataset: where the URoNet trained on one dataset is effectively used on another dataset (e.g., Table 3).
>
> We believe ImageNet-R has a different purpose than the work proposed in this research. In this work similar to the existing works provided by the reviewer, we aim to tackle the noises that are added to the images to fool the networks. Whereas, ImageNet-R has a different set of images belonging to categories such as art, toy, Embroidery, and Sculpture. The authors of ImageNet-R have also not used it for the robustness experiments but benchmarked it for out-of-distribution classification. CIFAR10-C and CIFAR100-C are the benchmark datasets for this purpose which we have already used in this study.
>
> *However, based on the suggestion of the reviewer, we have performed the robustness evaluation of the proposed URoNet against **auto attacks** [9]. The robustness of the proposed approach is 6.3% and 4.7% better than [6].*
>
> **The generalization of the proposed approach against any corruption and adversarial perturbation is its true functionality for the true world. On top of that, the proposed UroNet is agnostic to architecture, which is reflected through its performance on various models having different widths and depths.**
>
> **Response to General Comment:** Based on the suggestion of the reviewer, we have performed the comparison with the most recent method [6]. The comparison analysis has been reported in Table 6. It is to be noted here that the proposed URoNet does not utilize any knowledge of adversarial perturbation.
>
> The proposed algorithm has surpassed the existing algorithm [6] by a significant margin in handling the common corruption as well. For example, on Gaussian corruption, the performance of the proposed URoNet is at least 13.3% better than [6].
>
> Further, it is interesting to observe that the existing method is highly vulnerable to the depth of the model. For example, the SD + Reg [6] obtained a robust accuracy of 56.89% when the ResNet-9 model was used; however, the accuracy drastically dropped to 44.42% when a deeper version (ResNet-20) was used. Since the proposed UroNet does not utilize any knowledge of a target network or its size (it is a separate autoencoder network), is not sensitive to these changes.
>
> It is found that the frequency-based approaches are sensitive to the architecture. On top of that, the approach in [6] failed drastically when the attack was applied in the frequency domain. Whereas, the proposed approach does not utilize any knowledge of adversarial or target architecture.
>
> **An extensive comparison with several other state-of-the-art approaches can be found in Tables 6 to 9. The proposed URoNet surpasses the existing approaches by a significant margin, demonstrating its effectiveness in handling seen or unseen complex attacks.**
>
> [6] Improving native CNN robustness with filter frequency regularization. TMLR 2023.
>
> [9] Reliable evaluation of adversarial robustness with an ensemble of diverse parameter-free attacks. ICML 2020
>
> [A] Evaluating Adversarial Robustness in the Spatial Frequency Domain. arXiv 2024.

---

> > ### Author Response · Authors · 2024-12-29
> > **Response to Weaknesses**
> >
> > **Use of Noise in Degradation Detection Network:** The prime reason for adding the noise can be seen from the fact the noises are external signals to the images whether added intentionally or unintentionally, the absence of knowledge of such an external factor, makes the classification network vulnerable against them or a defense network not generalized against unseen noises. Therefore, the use of noise in the degradation detection network acts as providing subtle auxiliary information to regularize the detection in identifying perturbed images. We assert that such a supervisory signal helps the network better distinguish the clean samples from the noisy images. However, as demonstrated different noises (corruption or adversarial) share a similar effect through Figure 1 and the impact of frequency components through existing literature (Lukasik et al. (2023); Saikia et al. (2021); Yin et al. (2019b)), the inclusion of one type of noise can detect other types of noises.
> >
> > We have also performed an ablation study to showcase the impact of augmenting the noise as an auxiliary signal in detecting noisy images. The discussion can be found in section 3.3 under the “Impact of Augmenting Noise heading”.
> >
> > We have added the above information in the ablation study section mentioned above.
> >
> > **Evaluation on ViT:** It is observed that the ViTs are less sensitive to common corruption than CNNs [C,D]. Moreover, as mentioned the proposed approach is agnostic to the target classifier: only aims to provide a clean image to the target classifier after removing the noise, it shows a significant boost in the recognition performance of ViT as well.
> >
> > Table: mCEs (%) of different models and methods on ImageNet-C (lower is better).
> > | Model              | mCE  |
> > |--------------------|------|
> > | ResNet-50          | 76.7 |
> > | BiT m-r101x3       | 58.3 |
> > | DeepAugment+AugMix | 53.6 |
> > | ViT L-16           | 45.5 |
> > | Proposed           | **41.8** |
> >
> > **Notation:** f represents the input clean image, \eta represents the noise vector/image which can be any (common corruption or an adversarial perturbation). The modification of f through \eta results in an attacked image “g”. Since, the noise vector extracted from the noise remover network (part “A” of Figure 3) is of the same dimension as the input, therefore, after its concatenation with an image along the channel (c) dimension, the number of channels will be doubled which is mentioned through description and notation of M*N*2c. We have modified the notations based on the suggestion of reviewer Sb6p.
> >
> > Part A of the architecture outputs two images: one assumed to be a clean image and another assumed to be noise added in. Equation 2, reflects two mean square errors: one computed between the true noise and predicted noise image and another computed between the true clean image and predicted clean image. Once, part A is trained and the network can accurately predict the inherent noise, it has been added along the channel dimension to the images to train the degradation detection network.
> >
> > **Performance on Clean Images:** The experiments on clean images can be performed in the following two fashions:
> >
> > (i) Since part A of the network aims to learn the inherent noise, it predicted the black vector (since there is no noise information). We have not observed any reduction in the clean accuracy. (ii) Further, we can also understand this from the requirement of having a degradation detection network that reflects significant success in detecting corrupted images, hence, if the image is predicted clean, there is no need to pass it through a noise remover network and hence no reduction is observed in the clean image accuracy.
> >
> > Cost: The computation cost of the proposed approach is independent of the target classifier and is significantly lower than the existing approach which (re-) trains the target classifiers with regularization [6,B].

---

> ### Author Response · Authors · 2024-12-29
> **Responses to Remaining Points**
>
> **Recognition Accuracy with and without URoNet:** If we understood correctly then the reviewer is looking for the accuracy of the classifiers with and without the use of URoNet on corrupted images. The findings are reported in terms of the mitigation performance of the proposed approach (Section 3.2). For example, Figures 6 & 7 show the performance of backbones, ResNet (left) and Wide-ResNet16-8, and WRN28-10 on CIFAR datasets. Similarly, Tables 4 and 5 show the performance of other backbone architectures on Tiny ImageNet and ImageNet, respectively.
>
> **Grad-CAM:** We assert that th Grad-CAM analysis provides the support that once the proposed URoNet removes the noise from the images, the target classifier can focus on critical regions of images and hence might be a reason for improved performance. We agree on clean images, the Grad-CAM does not show that precision as observed on noisy images, however, it showcases the network focus on critical regions (e.g., columns 1, 3, and 5). To address this concern of the reviewer, we have now moved the Grad-CAM analysis to the appendix of the paper.
>
> Further, based on the suggestion of the reviewer, we have converted the values in terms of accuracy in place of improvements in Figures 6 and 7. Based on the suggestion of reviewer Sb6p, we have also carefully do the adjustment of the materials including the movement of Table 1 to the appendix. Further, we have condensed Tables 1 and 2 provided the average detection accuracy in the main paper, and moved the detailed Tables to the appendix (Tables B2 and B3). Similarly, Tables 7 and 8 (now Tables C6 and C7) are compressed to the key results and replaced with Figures 6 and 7.
>
> [A] Liao KH, Yeh CY, Chen HW, Chen MS. Evaluating Adversarial Robustness in the Spatial Frequency Domain. arXiv preprint arXiv:2405.06345. 2024.
>
> [B] Guo Y, Stutz D, Schiele B. Improving robustness of vision transformers by reducing sensitivity to patch corruptions. In Proceedings of the IEEE/CVF Conference on Computer Vision and Pattern Recognition 2023 (pp. 4108-4118).
>
> [C] Morrison K, Gilby B, Lipchak C, Mattioli A, Kovashka A. Exploring corruption robustness: Inductive biases in vision transformers and MLP-mixers. arXiv preprint arXiv:2106.13122. 2021 Jun 24.
>
> [D] Paul S, Chen PY. Vision transformers are robust learners. In Proceedings of the AAAI conference on Artificial Intelligence 2022 Jun 28 (Vol. 36, No. 2, pp. 2071-2081).

---

> ### Comment · Reviewer_ijzf · 2025-01-07
> **Thank you for the response**
>
> I thank the authors for their response and appreciate the changes made in the paper.
> However, there are still some unanswered questions that I would like to address:
>
> - Table 6: I thank the authors for including this comparison. However, the comparison appears to be somewhat inequitable. Most of the methods use various forms of distillation with different training approaches, whereas the method [6] does not.
>
> - Additionally, I would be interested to see how these methods perform on common corruptions.
>
> - Regarding the discussion on Vision Transformers (ViTs), the statement, "It is observed that the ViTs are less sensitive to common corruption than CNNs," is well-known. This further emphasizes the importance of including ViTs in the analysis, as it would enhance the significance and applicability of the proposed method. The authors state that the work functions as "The generalization of the proposed approach against any corruption and adversarial perturbation is its true functionality for the real world." Moreover, the proposed UroNet is stated to be architecture-agnostic, which is reflected in its performance across various models with different widths and depths. This claim would benefit from evaluations of recent state-of-the-art methods, as it appears that some important networks are missing from the discussion. An inclusion of this would be great to have in the paper.
>
> - As previously mentioned, how would the proposed method perform on the cue-conflict dataset?
>
> - When comparing Table 1 and Table 2, why does uniform noise yield the best results for CIFAR-10, while speckle noise performs best for CIFAR-100? Is there an intuitive explanation for this?
>
> - In Section 3.2, regarding mitigation strategies, how does this work compare with other approaches for addressing common corruptions?
>
> - I also wonder why different CNNs were used for experiments on CIFAR-10 and CIFAR-100 (see Figures 6 and 7).
>
> - I would appreciate a discussion on the differences between this work and DutaNet.
>
> Overall, some of my points were satisfactorily addressed. However, I still feel that this paper lacks a comprehensive comparison with state-of-the-art techniques aimed at enhancing both adversarial robustness and addressing common corruptions jointly. Additionally, the proposed approach and the related methods presented throughout the work seem to be intermingled, making it difficult to evaluate their effectiveness individually. A more structured analysis that clearly delineates the proposed method and compares them against the existing approaches for both degradations mitigation would enhance the clarity and impact of the findings.
>
>
> *Minor points*:
> - In Figure 5, should it be  "IFGSM40" instead of "FGSM40" in the CIFAR-10 plot?
> - It would be beneficial to specify the network utilized in Section 3.1.
> - Additionally, a review for consistency in active and passive references would be advisable.

---

> > ### Author Response · Authors · 2025-01-17
> > **Response**
> >
> > **Comparison:** Thanks for pointing out this observation. We have updated the existing algorithm discussion to highlight this point. We believe that since the paper [6] aims to provide robustness to adversarial and corruption attacks, it might be an ideal fit for the comparison; however, the description can advance the readers of the paper on how this technique is different from other compared techniques.
> >
> > The work proposed in [6] also aims to provide robsutness against common corruptions, **we have checked its performance on the CIFAR10-C dataset and observed that the performance of the proposed model is at least 14% better than the SD + Reg.**
> >
> > [6] Jovita Lukasik, Paul Gavrikov, Janis Keuper, and Margret Keuper. Improving native CNN robustness with filter frequency regularization. TMLR 2023.
> >
> > *Based on the suggestion of the reviewer, we have further comprehensively expanded the comparison by performing the comparison on various ViT designs on the full ImageNet corruption dataset. The results are reported in Figure 8.*
> >
> > **Out-of-Distribution Datasets:** As mentioned earlier, ImageNet-R form data do not lie in the scope of this paper. However, as the dataset is developed for out-of-distribution, **we treat this as the case the train our degradation detection network for out-of-distribution detection instead of classifying an image as noisy or clean. Due to drastic distribution shifts (texture and style manipulation), the network can achieve close to perfect OOD detection performance.**
> >
> > *Furthermore, as the authors [1] have pointed out the “robustness intervention on ImageNet-C also helps with real-world blurry images. Hence ImageNet-C can track performance on real-world corruptions.”* **Therefore, the robustness of the proposed approach on ImageNet-C (Reported in Figure 8) demonstrates its effectiveness in handling real-world corrutions.**
> >
> > [1] The Many Faces of Robustness: A Critical Analysis of Out-of-Distribution Generalization
> >
> > We want to mention that in the future, we would like to address these forms of robustness as well which might be a limitation of the current work, and aim to extend the proposed URoNet to address this robustness issue. We have updated the conclusion section to highlight it.
> >
> >
> > While there is no ideal answer to this question; however, we believe that the prime difference between uniform and speckle noise can be a reason behind such observation. The difference between applying these two different noises such that speckle noise is a multiplicative noise that appears as random variations; whereas, uniform noise is an additive noise uniformly distributed. Speckle noise distorts some specific regions especially regions in brighter areas instead of uniformly distorting the image. We believe that of additive nature and uniform distribution make the Uniform noise an ideal choice to mitigate adversarial attacks. Figure 10 demonstrates that UN outperforms SPKN noise significantly. In the future, we further aim to understand this observation to provide improved robustness against corruption, adversarial noises, and cue-conflict datasets.
> >
> >
> > **Protocol:** The comparison has been performed using the pre-defined protocols. For example, CIFAR10-C and CIFAR100-C have already defined corrupted test sets. The purified images are generated using the proposed method on the pre-defined test sets and used to report the accuracy and mCE values. Further, the comparison with other methods such as those mentioned in Table 4 (Salman et al. (2020); Xie et al. (2019)) have been used to remove the individual noises, and cleaned images have been provided for classification and the accuracy have been compared with the proposed architecture. Table 5 and Table 6 follow the pre-defined test protocol so that fair comparisons can be made.
> >
> > Apart from that, an extensive set of comparisons with corruption-related defense algorithms, adversarial noise mitigation algorithms, unified defense algorithms, and a new comprehensive comparison including ViTs and adversarial training has been provided in the paper by following the same protocol used to evaluate the proposed model.
> >
> > **Models:** While one model is common between the CIFAR-10 and CIFAR-100 datasets, the reason for choosing the other bigger model is its better accuracy on clean images as compared to the shallow model such as ResNet. However, when we used the ResNet model on the CIFAR100 dataset it showed similar corruption mitigation performance as observed on any model on CIFAR100 and the performance observed of ResNet on the CIFAR10 dataset.
> >
> > Minor:
> >
> > 1. We have updated the typo in Figure 5.
> > 2. We want to mention that section 2.3 specifies the name of the network that we have used the Wide-ResNet-2.
> > 3. We have further proofread the paper to maintain consistency in active and passive references.
> >
> > We have restructured the comparison section so that it can be easily followed in continuation of the performance of the proposed algorithm on individual noises.

---

### Review · Reviewer_PV8Z · 2025-01-08

**Summary Of Contributions:**

This paper proposes URoNet, a framework that incorporates both natural and adversarial perturbations. The authors combine
noise remover network, image classification network, and degradation detection network to detect noises. By learning the connection between images and noises, URoNet improves the robustness under perturbations.

**Audience:**

Yes

**Claims And Evidence:**

Yes

**Requested Changes:**

See Weaknesses and Questions.

**Strengths And Weaknesses:**

Strengths:
1. This work combines the learning of common corruptions and adversarial noises in a unified framework.

2. Experimental results are across various datasets and noise severities.

Weaknesses:
1. Recent SOTA robust methods show their robustness in the entire ImageNet dataset. For a thorough comparison with those works, results on (entire) ImageNet should be given.

2. Baselines are not sufficiently discussed. For example, GLOT-DR [1] shows 83.7% accuracy in CIFAR-10-C (which is better than URoNet).

3. There is an inevitable trade-off between the accuracy under clean images and robustness. I recommend the authors to include the experimental analysis of such trade-off.

Questions:
1. What is f(x,y) in p4? It seems x, y are not defined.

2. Are noise removal network, image classification network, degradation detection network trained simultaneously? And does the entire training require 25min with 2080 GPU?

3. Salman et al., [2] (in Table 7) provides certified robustness (instead of empirical robustness). How are the scores in Table 7 calculated? Is it certified accuracy?

[1] Phan et al., Global-Local Regularization Via Distributional Robustness, ICML 2023

---

> ### Author Response · Authors · 2025-01-17
> **Rebuttal**
>
> **Weakness Responses:**
>
> **Comparison and Full ImageNet:** Based on the suggestion of the reviewer, we have added the comparison on the ImageNet-C dataset in Figure 8. It can be seen that the proposed algorithm outperforms the complex vision transformer architecture containing millions of trainable parameters by a significant margin except in the case of DeiT-B + cond ANT which yields 40.68 mCE compared to 41.8 mCE of the proposed URoNet. It is to be noted here that the number of parameters of DeiT-B is 86M which is drastically higher than the proposed method and DeiT-B + cond ANT utilizes heavy adversarial noise training (ANT) which further increases its compilation cost. We have also performed the comparison with another recent algorithm ShCA Zhang et al. (2024), which also utilizes the augmentation strategy but in a way of selecting semi-hard triplet samples.
>
> **Baseline:** While we agree with the observation that the performance of GLOT-DR is better than URoNet, we firmly believe that the difference is not significant. Furthermore, the performance of URoNet (63.5) is drastically better than GLOT-DR (55.7) on the CIFAR100-C datasets. We have updated the comparison in Table 5. Apart from that, based on the suggestion, extensive comparisons on ImageNet-C are further added in the paper.
>
>
> **Trade-off:** Performance on Clean Images: As pointed to the *reviewer ijzf*, the trade-off between the clean accuracy and robustness handled in the following two fashions:  **(i)** Since part A of the network aims to learn the inherent noise, it predicted the black vector (since there is no noise information). Henceforth, we have not observed any reduction in the clean accuracy. **(ii)** Further, we can also understand this from the requirement of having a degradation detection network that reflects significant success in detecting corrupted images, hence, if the image is predicted clean, there is no need to pass it through a noise remover network and hence no reduction is observed in the clean image accuracy.
>
> **Response to Questions:**
>
> 1. Based on the suggestion of the reviewers, we have earlier modified the notations of the equations and hence no (x,y) component which were earlier denoting the pixel coordinates of a 2D image. However, to address the concerns of reviewer Sb6p, we have updated the notations in the paper.
>
> 2. Since the degradation detection network requires the noise vector computed from the image remover network and the classification network requires clean images; henceforth, in the proposed setting, the networks are trained individually. Only the networks that need to be trained are noise remover and degradation detection. Further, the exclusion of the image classification network can be seen from the fact that the proposed contribution lies in the development of a noise remover network and degradation detection network which can act as a plug-in to any image classification model. The effectiveness of the proposed noise removal network and degradation detection has been extensively studied on multiple architectures.
>
> We have updated the sentence to avoid any confusion.
>
> 3. We have trained the image-denoising network based on the strategy proposed by Salman et al.. The authors have proposed denoised smoothing that performs randomized smoothing utilizing two different metrics: the mean squared error objective (MSE), and the stability objective. The smoothed/denoise images are later used for classification.
>
> [1] Phan et al., Global-Local Regularization Via Distributional Robustness, ICML 2023

---

### Decision · Action_Editor_mZ1g · 2025-02-26

**Recommendation:** Reject

**Comment:**

The authors did a decent job of trying to address the reviewers concerns, and there was agreement that the paper was much improved post-rebuttal. However, the reviewers still felt that there were some key improvements missing to back up the claims, most notably: (1) inclusion of a thorough sample of existing approaches that also tackle both types of degradation (common corruptions and adversarial perturbations), (2) more thorough exploration of how the framework could help ViTs, (3) using the same networks for CIFAR-10 and CIFAR-100 tests, (4) a proper discussion about the differences between this work and DuTaNet, which is closely related, and (5) clear writing that is more consistent in terminology and highlights the motivations more fully.

**Audience:**

This paper would be of interest to some members of the TMLR community, specifically those interested in dealing with image recognition in the presence of image degradations. However, it is not very novel, which would limit its impact. Nonetheless, given that novelty is not supposed to be a criteria for acceptance, this consideration did not factor into the decision.

**Claims And Evidence:**

This paper proposes a framework, called URoNET (the acronym is not explained), for detecting different types of image degradation and improving the accuracy of models in image categorization when presented with degraded images. The degradations consider include common noise corruptions and adversarial perturbations. In the framework, after noise is detected, a clean image is generated, and is then be used for classification by another downstream network. The entire structure can be trained end-to-end.

The authors claim that their framework is effective for detecting degredations and mitigating the impact of corruption to enhance recognition performance. The method was evaluated on several datasets, including CIFAR-10, CIFAR-100, and TinyImageNet. The reviewers raised some concerns that the evaluations did not include enough comparisons to other techniques for dealing with image perturbations to support the claims, and that there was lack of clarity in the manuscript about the motivations.

**Resubmission Of Major Revision:**

The authors may consider submitting a major revision at a later time.